# VidBridge-R1: Bridging QA and Captioning for RL-based Video Understanding Models with Intermediate Proxy Tasks

**Xinlong Chen**[1,2,3,*] **Yuanxing Zhang**[3], **Yushuo Guan**[3], **Weihong Lin**[3], **Zekun Wang**[3],

**Bohan Zeng**[4], **Yang Shi**[4], **Sihan Yang**[4], **Qiang Liu**[1,2,†], **Pengfei Wan**[3], **Liang Wang**[1,2]

[1]New Laboratory of Pattern Recognition (NLPR),
Institute of Automation, Chinese Academy of Sciences (CASIA)
[2]School of Artificial Intelligence, University of Chinese Academy of Sciences
[3]Kling Team, Kuaishou Technology    [4]Peking University

## Abstract

The "Reason-Then-Respond" paradigm, enhanced by Reinforcement Learning, has shown great promise in advancing Multimodal Large Language Models. However, its application to the video domain has led to specialized models that excel at either question answering (QA) or captioning tasks, but struggle to master both. Naively combining reward signals from these tasks results in mutual performance degradation, which we attribute to a conflict between their opposing task natures. To address this challenge, we propose a novel training framework built upon two intermediate proxy tasks: *DarkEventInfer*, which presents videos with masked event segments, requiring models to infer the obscured content based on contextual video cues; and *MixVidQA*, which presents interleaved video sequences composed of two distinct clips, challenging models to isolate and reason about one while disregarding the other. These proxy tasks compel the model to simultaneously develop both holistic, divergent understanding and precise, convergent reasoning capabilities. Embodying this framework, we present VidBridge-R1, the first versatile video reasoning model that effectively bridges the paradigm conflict. Extensive experiments show that VidBridge-R1 achieves significant performance gains on both QA and captioning within one model, demonstrating the efficacy of our approach in fostering more generalizable and powerful video understanding models. Code is available at https://github.com/VidBridge-R1/VidBridge-R1.

## 1 Introduction

The release of OpenAI o1/o3 (Jaech et al., 2024) and DeepSeek-R1 (Guo et al., 2025) has introduced a novel *Reason-Then-Respond* paradigm to the development of large language models (LLMs), which significantly enhances model performance through test-time scaling. Inspired by this approach, a growing body of research (Team et al., 2025; Chen et al., 2025a; Shen et al., 2025; Deng et al., 2025; Xia et al., 2025; Yao et al., 2025) has extended this paradigm to multimodal large language models (MLLMs). By leveraging reinforcement learning (RL), particularly the Group Relative Policy Optimization (GRPO) algorithm (Shao et al., 2024), these studies have achieved promising results in image-based reasoning tasks.

Recently, several studies (Feng et al., 2025; Zhang et al., 2025c; Chen et al., 2025g;h) have begun to explore the application of the *Reason-Then-Respond* paradigm in the video modality. Some efforts focus on enhancing question answering (QA) capabilities in general or reasoning scenarios (Li et al., 2025c; Dang et al., 2025), while some other works concentrate solely on improving video captioning performance (Li et al., 2025b; Meng et al., 2025a). However, these approaches remain narrowly

---

*This work was conducted during the author's internship at Kling Team, Kuaishou Technology

†Corresponding author: qiang.liu@nlpr.ia.ac.cn

tailored to specific tasks and fail to achieve an effective integration of both QA and captioning within a unified model framework.

A key advantage of MLLMs lies in their versatility, enabling strong performance across diverse tasks simultaneously. It is therefore undesirable to enhance reasoning capabilities at the expense of generalizability by over-specializing the model in a single task. To preserve generality in both QA and captioning tasks, an intuitive approach is to combine the reward signals from them during training. However, as shown in Figure 1, simply mixing the datasets and rewards leads to performance degradation on both tasks. We attribute this phenomenon to a conflict in the learning paradigms between QA and captioning tasks when training with RL, an indirect optimization method guided by global reward signals. Specifically, QA is inherently a convergent reasoning task that requires the model to locate and output a unique, low-entropy correct answer from abundant information. In contrast, captioning is a divergent generation task that demands the model to produce diverse and comprehensive high-entropy descriptions. These opposition task objectives may lead to an inherent conflict during RL optimization.

To solve this problem, we resort to two proxy tasks that focuses more on comprehensive understanding of the videos, rather than focusing on the downstream task form. Inspired by the "fill-in-the-middle" (Guo et al., 2024) in code generation and "style mixing (Karras et al., 2019)" in image generation, we design Dark-EventInfer and MixVidQA, aiming to bridge the paradigm gap between QA and captioning. Specifically, in DarkEventInfer, certain event segments within the original videos are masked out with black screens, and the model is required to deduce and predict the masked events based on contextual video cues. In MixVidQA, two distinct videos are interleaved, with questions targeting only one of them, and the model must identify and reason about the most rele-

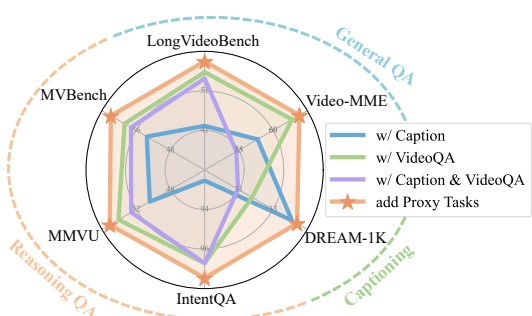

Figure 1: Performance comparison on QA and captioning tasks under different training setups. Details can be found in Section 6.3

vant video content to provide accurate answers. These tasks compel the model to perform both divergent holistic understanding of the video context (as required in captioning) and convergent pinpointing of key information (as needed in QA), thereby enhancing structured video representation and contextual reasoning abilities. Incorporating these proxy tasks during training, we present VidBridge-R1, a versatile video reasoning model that excels at answering questions in general or reasoning scenes, as well as video captioning. As illustrated in Figure 1, VidBridge-R1 effectively alleviates the paradigm conflict between QA and captioning, leading to significant performance gains in corresponding tasks. Our contributions can be summarized as follows:

- We develop VidBridge-R1, the first versatile video understanding model capable of simultaneously handling QA and captioning tasks under the Reason-Then-Respond paradigm.
- We propose two novel intermediate proxy tasks, DarkEventInfer and MixVidQA, designed to bridge the paradigm gap between the divergent nature of video captioning and convergent reasoning demands of question answering in general or reasoning scenarios.
- Extensive experiments demonstrate that VidBridge-R1 achieves significant performance improvements on various video general understanding, cognitive reasoning, and video captioning tasks.

## 2 RELATED WORK

### 2.1 MULTIMODAL UNDERSTANDING MODELS

Multimodal understanding models are widely recognized as a crucial step toward achieving artificial general intelligence (AGI) and have seen remarkable progress in recent years (Zhang et al., 2025a; 2026; Shi et al., 2025a;c; Chen et al., 2025d; 2026; Hua et al., 2025). The LLaVA series (Liu et al., 2023) aligns visual and language representations through fully connected layers, equipping LLMs with the ability to interpret visual inputs. The Intern-VL series (Chen et al., 2024b) achieves more

sophisticated visual understanding by employing a large-scale visual encoder to capture fine-grained details. Beyond architectural innovations, various methodological approaches have been proposed to improve the model's multimodal understanding abilities. To process longer video sequences, some studies (Li et al., 2024b; Shu et al., 2024) compress visual information, while others (Wang et al., 2024d; Chen et al., 2024a) extend the context window of LLMs, enabling the modeling of longer temporal sequences of video frames. Additionally, specialized models like Tarsier2 (Yuan et al., 2025), Mavors (Shi et al., 2025b) and CogVLM2-Caption (Hong et al., 2024) have shown impressive results in video captioning through carefully designed training pipelines and diverse datasets. However, these models remain limited to conventional video understanding paradigms. In this work, we aim to move beyond traditional frameworks and explore the performance gains achieved through the *Reason-Then-Respond* paradigm.

## 2.2 MULTIMODAL REASONING MODELS

Following the success of DeepSeek-R1 (Guo et al., 2025), research on multimodal reasoning models has experienced rapid advancement. In the realm of image-based reasoning, numerous studies (Huang et al., 2025; Peng et al., 2025b; Tan et al., 2025; Chen et al., 2025b;c) achieve reasoning and reflection capabilities by training models on geometry-related datasets. MM-Eureka (Meng et al., 2025b), on the other hand, compiles a wide range of problems from education curricula to simulate human learning processes, thus improving the model's cross-disciplinary reasoning abilities. Additionally, other efforts (Liu et al., 2025c;b; Shen et al., 2025) focus on improving object detection and grounding capabilities by designing Intersection-over-Union (IoU) related reward functions. In the realm of video-based reasoning, Video-R1 (Feng et al., 2025) and VideoRFT (Wang et al., 2025) adopt a two-stage training strategy that combines SFT with RL, enhancing the model's QA capabilities in general or reasoning scenarios after training on extensive image and video datasets. VideoChat-R1 (Li et al., 2025b), in contrast, applies RL directly to develop task-specific reasoning models. Meanwhile, VideoCap-R1 (Meng et al., 2025a) focuses exclusively on advancing video captioning performance. Despite these efforts, existing methods remain narrowly tailored to particular tasks and lack an effective integration of QA and captioning within a unified model framework. Our work aims to bridge this gap by leveraging intermediate proxy tasks.

## 3 PROBLEM FORMULATION

The powerful versatility of MLLMs is one of the key reasons for their popularity and widespread adoption. For video understanding models, important application scenarios include perceptual understanding of general scenes, cognitive reasoning in complex scenarios, and the generation of fine-grained captions for video content. However, recent studies adopting the *Reason-Then-Respond* paradigm have often improved model performance in specific application scenarios at the expense of the model's overall versatility. In this study, we aim to enhance the model's generalist capabilities across the aforementioned application scenarios by introducing intermediate proxy tasks to bridge the diverse capability requirements imposed by different applications. The complete implementation details, including the data curation pipeline and the reward function specifications, are presented in subsequent sections of this work.

## 4 DATA CURATION

In this section, we present the curation process of the training data, including the two intermediate proxy tasks designed to bridge the gap between QA capabilities in general or reasoning scenarios and video captioning tasks. By jointly training on conventional VideoQA and captioning tasks along with our proposed proxy tasks, the model not only learns the basic output format but is also encouraged to integrate divergent holistic understanding of the video content (as required in captioning) with convergent pinpointing of key information (essential for QA). This training approach promotes structured video representation and enhances contextual reasoning abilities, thereby mitigating the divergence between QA and captioning while improving performance on both. Illustrative examples of from each task can be found in the right part of Figure 2 or in Appendix B.3.

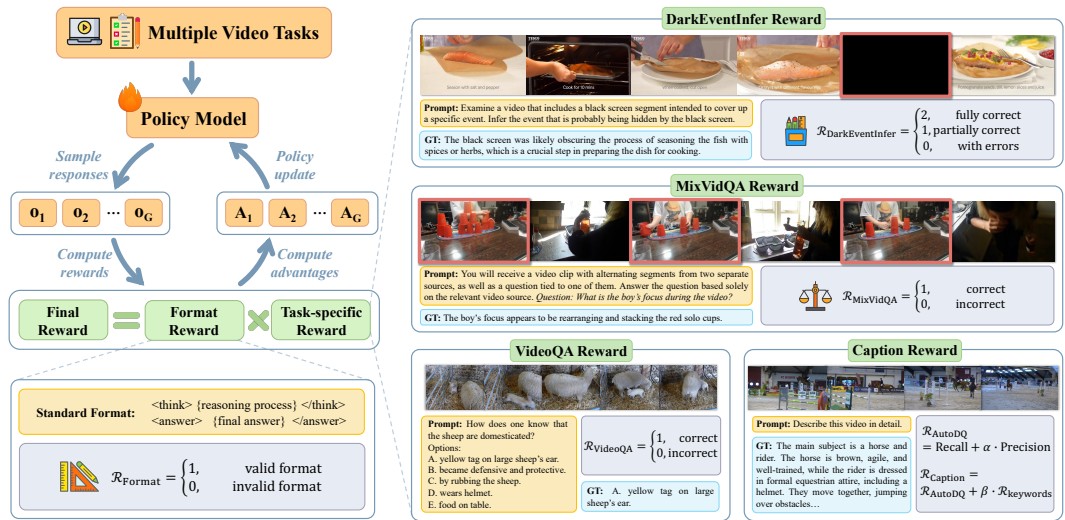

Figure 2: The training framework of VidBridge-R1. By incorporating intermediate proxy tasks, VidBridge-R1 effectively bridges the gap between QA capabilities in general or reasoning scenarios and video captioning tasks.

## 4.1 CURATION OF DARKEVENTINFER

Inspired by the "fill-in-the-middle" (Guo et al., 2024) approach in code generation, we design the first proxy task, DarkEventInfer, to encourage holistic video context understanding. Specifically, in this task, the model is presented with a video containing black-screen segments and is required to infer the events that occur during the masked periods based on contextual video cues.

Using event captions and their corresponding timestamp annotations from COIN (Tang et al., 2019), we randomly select one event per video and replace it with a black screen. The model is then asked to reason about and predict the masked event based on the contextual information provided by the surrounding visible segments.

To ensure the quality and learnability of the dataset, we conduct a human evaluation on the masked videos. Instances where human annotators are unable to infer the masked event are removed, and any inaccurate or ambiguous captions are revised. These steps ensure the final dataset offers meaningful contextual signals, enabling models to effectively learn how to reason about the "dark events".

## 4.2 CURATION OF MIXVIDQA

Drawing inspiration from the "style mixing" (Karras et al., 2019) technique in image generation, we design the second proxy task, MixVidQA, to enhance the model's ability to attend to and extract key information from videos. In this task, the model is presented with interleaved video sequences from two distinct video clips, and is required to answer questions based on the most relevant video while ignoring the other.

We source video clips from Kinetics (Kay et al., 2017), each lasting around 10 seconds. To construct the mixed video sequences, we randomly select two video clips and interleave them with a random interval ranging from 1.5 to 2 seconds. For each mixed video, we generate a set of QA pairs using Qwen2-VL-72B (Wang et al., 2024b), explicitly referring to one of the original clips. The model to be trained is expected to identify the relevant video segment to provide the correct answer based on the mixed video sequences.

All generated QA pairs undergo manual review to ensure quality. Any pairs with ambiguous references or unclear ground truths are removed, guaranteeing the reliability of the dataset.

### 4.3 CURATION OF VIDEOQA AND CAPTIONING DATA

In addition to the two proxy tasks described above, we incorporate additional data for conventional VideoQA and video captioning to explicitly familiarize the model with the target output formats of these downstream tasks. For the conventional VideoQA task, we sample instances from the training set of NExT-QA (Xiao et al., 2021), ensuring broad coverage of question types. For the captioning task, which demands richer linguistic generation, we curate high-quality videos from Koala-36M (Wang et al., 2024c) and KVQ (Lu et al., 2024), prioritizing video diversity in aspect ratios, motion intensity, visual scenes, and thematic content. This enables the model to gain a deeper understanding of dynamic video content. For Koala-36M, we adopt the original captions. For KVQ that lacks video captions, we employ Gemini-2.5-Pro (Gemini Team, 2025) to generate high-quality textual descriptions.

### 4.4 DATA FILTERING FOR GRPO TRAINING

Considering that the GRPO algorithm may become ineffective when all candidate answers receive identical rewards, resulting in zero advantage functions, we adopt a pre-filtering strategy. Specifically, we utilize Qwen2.5-VL-7B (Bai et al., 2025), forcing it to engage in reasoning by specific prompt and sampling five responses with a temperature of 1.0. For the captioning task, we compute the F1 score of each response using AutoDQ (Wang et al., 2024a), and discard samples where the variance of the F1 scores across the five responses is less than 0.2. For other tasks, we filter out questions for which all generated answers are uniformly correct. The above methods ensure effective policy updates during GRPO training. The final training set comprises 1,841 samples from DarkEventInfer, 2,332 from MixVidQA, 2,003 from VideoQA, and 4,624 from the Captioning task, totaling 10,800 high-quality training instances.

## 5 TRAINING STRATEGY

Many existing works (Feng et al., 2025; Zhang et al., 2025b) adopt a two-stage training approach, where the model is first trained via SFT and then refined by RL. However, we find that when using high-quality reasoning data with carefully designed intermediate proxy tasks, the SFT stage is not only unnecessary but can also impair the model's inherent reasoning potential by forcing it to learn a specific reasoning pattern (as illustrated in Appendix E). In contrast, applying RL directly to the model can effectively stimulate its reasoning abilities. For the RL algorithm, we employ GRPO without KL divergence regularization (i.e., setting $\lambda = 0$ in Equation 2, analysis can be found in Appendix C.3), and design diverse reward functions for different tasks, as shown in Figure 2 and detailed in the following.

### 5.1 GROUP RELATIVE POLICY OPTIMIZATION

Group Relative Policy Optimization (GRPO) significantly reduces both training time and GPU memory usage by eliminating the need for a separate critic model in Proximal Policy Optimization (PPO). Specifically, GRPO works by sampling a group of $G$ responses $\{o_1, o_2, ..., o_G\}$ for each question $q$ from the old policy model $\pi_{\theta_{old}}$, then computing their corresponding rewards $\{r_1, r_2, ..., r_G\}$ to derive the advantage function $A_i$ for response $o_i$:

$$A_i = \frac{r_i - \text{mean}(\{r_1, r_2, \ldots, r_G\})}{\text{std}(\{r_1, r_2, \ldots, r_G\})} \tag{1}$$

The current policy model $\pi_\theta$ is then optimized using the following objective function:

$$\begin{aligned} \mathcal{J}_{\text{GRPO}}(\theta) = \mathbb{E}_{\{o_i\}_{i=1}^G \sim \pi_{\theta_{\text{old}}}(o_i|q)} &\left[ \frac{1}{G} \sum_{i=1}^G \left( \min\left( \frac{\pi_\theta(o_i|q)}{\pi_{\theta_{\text{old}}}(o_i|q)} A_i, \right. \right. \right. \\ &\left. \left. \left. \text{clip}\left( \frac{\pi_\theta(o_i|q)}{\pi_{\theta_{\text{old}}}(o_i|q)}, 1 - \varepsilon, 1 + \varepsilon \right) A_i \right) - \lambda \cdot \mathbb{D}_{\text{KL}}\left( \pi_\theta || \pi_{\text{ref}} \right) \right) \right], \end{aligned} \tag{2}$$

## 5.2 REWARD FUNCTION DESIGN

### 5.2.1 DARKEVENTINFER

For the DarkEventInfer task, we use Qwen2.5-72B (Yang et al., 2024) as the judge to assess the quality of model responses. Given the inherent difficulty in accurately describing black-screen events, the judge model is prompted to make three-tier evaluations: fully correct, partially correct but error-free (i.e. incomplete but without introducing incorrect claims), or containing any errors, with rewards of 2, 1, and 0, respectively, as formalized in Equation 3.

$$\mathcal{R}_{\text{DarkEventInfer}} = \begin{cases} 2, & \text{if the answer is fully correct} \\ 1, & \text{if partially correct and error-free} \\ 0, & \text{if the answer contains any errors} \end{cases} \tag{3}$$

### 5.2.2 MIXVIDQA

For the MixVidQA task, we also employ Qwen2.5-72B as the judge. Empirical observations reveal that QA tasks are generally less challenging than accurately describing black-screen events. Therefore, a two-tier evaluation scheme is implemented for MixVidQA: a correct answer receives a reward of 1, while an incorrect one receives 0.

### 5.2.3 VIDEOQA

For the conventional VideoQA tasks, particularly multiple-choice questions, the selected options are extracted from the model's output using regular expressions and compared with the ground truth. A reward of 1 is assigned for correct answers, and 0 for incorrect ones.

$$\mathcal{R}_{\text{MixVidQA}} = \mathcal{R}_{\text{VideoQA}} = \begin{cases} 1, & \text{if the answer is correct} \\ 0, & \text{if the answer is incorrect} \end{cases} \tag{4}$$

### 5.2.4 CAPTIONING

In the captioning task, GPT-3.5-Turbo[1] is utilized as the judge model (analysis can be found in Appendix C.1) and the AutoDQ (Wang et al., 2024a) methodology is employed to calculate the event-level recall and precision of model-generated captions relative to ground-truth captions. The weighted sum of the two metrics constitutes the AutoDQ reward $\mathcal{R}_{\text{AutoDQ}}$:

$$\mathcal{R}_{\text{AutoDQ}} = \text{Recall} + \alpha \cdot \text{Precision} \tag{5}$$

Here, $\alpha$ is a weighting factor set to 0.5, which balances the contributions of recall and precision. Given that precision can be enhanced by generating more concise captions, while recall necessitates more comprehensive descriptions of video content, this setting helps equalize their improvement difficulty and mitigates the risk of reward hacking.

Moreover, we design a keywords reward $\mathcal{R}_{\text{keywords}}$ based on two predefined keyword sets: a temporally relevant set $T$ and a speculation-related set $S$, which aims to promote the inclusion of temporal keywords in the generated captions $C$ while discouraging speculative or irrelevant content. Examples of these keywords are provided in Appendix B.4.

$$\mathcal{R}_{\text{keywords}} = \begin{cases} -\sum_{w \in C} \mathbb{I}(w \in S), & \text{if } \exists w \in C \wedge w \in S \\ \min\left(\sum_{w \in C} \mathbb{I}(w \in T), \gamma\right), & \text{otherwise} \end{cases} \tag{6}$$

Here, $\gamma$ serves as the upper bound for temporal keyword rewards, empirically set to 2, to prevent excessive meaningless temporal keywords. The final caption reward $\mathcal{R}_{\text{Caption}}$ combines both components through a weighted summation:

$$\mathcal{R}_{\text{Caption}} = \mathcal{R}_{\text{AutoDQ}} + \beta \cdot \mathcal{R}_{\text{keywords}} \tag{7}$$

The weighting coefficient $\beta$ is set to 0.2 here, reflecting the secondary role of the keywords reward compared to event recall and precision. Further discussion of hyperparameter choices is provided in Appendix C.2.

---

[1] https://platform.openai.com/docs/models/gpt-3.5-turbo

### 5.2.5 FORMAT REWARD

Additionally, we introduce a format reward to encourage structured outputs that follow the *Reason-Then-Respond* paradigm. The reward is formally defined as:

$$\mathcal{R}_{\text{format}} = \begin{cases} 1, & \text{valid format} \\ 0, & \text{invalid format} \end{cases} \tag{8}$$

Unlike existing approaches that often treat format compliance as a separate reward component, we implement it implicitly. Specifically, only responses that conform to the required format are eligible to receive any of the task-specific rewards, as formalized in Equation 9. This design serves as a critical mechanism to prevent two common forms of reward hacking observed in prior work: (a) correct formats but incorrect answers; and (b) correct answers but incorrect formats. By enforcing format compliance as a prerequisite of task-specific rewards, our training framework could accelerate the convergence toward outputs that are both structurally correct and semantically accurate.

$$\mathcal{R}_{\text{total}} = \mathcal{R}_{\text{format}} \cdot \left( \mathcal{R}_{\text{DarkEventInfer}} + \mathcal{R}_{\text{MixVidQA}} + \mathcal{R}_{\text{VideoQA}} + \mathcal{R}_{\text{Caption}} \right) \tag{9}$$

## 6 EXPERIMENTS

### 6.1 EXPERIMENTAL SETTINGS

#### 6.1.1 BENCHMARKS

For QA tasks in general video understanding, we conduct experiments on Video-MME (Fu et al., 2024), LongVideoBench (Wu et al., 2024) and MVBench (Li et al., 2024a). For video reasoning tasks, we extend experiments on MMVU (Zhao et al., 2025), NExT-QA (Xiao et al., 2021), IntentQA (Li et al., 2023), Causal-VidQA (Zang et al., 2023), Video-Holmes (Cheng et al., 2025), as well as our held-out test sets: DarkEventInfer-Test and MixVidQA-Test. Each of the latter two contains 100 test samples. For video captioning tasks, we evaluate models on DREAM-1K (Wang et al., 2024a) and VidCapBench (Chen et al., 2025f).

#### 6.1.2 BASELINES

First, we evaluate both the direct outputs and the reasoning-elicited responses of Qwen2.5-VL-7B-Instruct (Bai et al., 2025) as fundamental baselines. For contemporaneous works, we conduct comparative analyses with three prominent models: Video-R1 (Feng et al., 2025), VideoChat-R1 (Li et al., 2025b), and VideoRFT (Wang et al., 2025). Additionally, we perform SFT on Qwen2.5-VL-7B using our constructed 10k training samples for 3 epochs and include it in the comparison.

#### 6.1.3 IMPLEMENTATION DETAILS

We employ Qwen2.5-VL-7B-Instruct as the backbone model for training. To ensure training efficiency and stability, we uniformly sample 16 frames from each video at the maximum resolution of $196 \times 28 \times 28$. For GRPO training, we sample 8 responses per question with a temperature of 1.0 to ensure diversity. The learning rate is set to 1e-6, and the batch size is 32. During inference, for QA tasks, video frames are sampled at 1 fps with a maximum of 128 frames per video, while maintaining the same resolution as in training. For captioning tasks requiring more details, we uniformly sample 16 frames from each video, which in most cases corresponds to a sampling rate exceeding 2 fps. Greedy decoding is employed during inference to ensure deterministic outputs, and the output length is limited to 2,048. All experiments are conducted on 8 NVIDIA A800 GPUs.

### 6.2 EXPERIMENTAL RESULTS

We present the performance comparison of VidBridge-R1 with baseline models on general video understanding and captioning tasks in Table 1, and on video reasoning tasks in Table 2.

In QA tasks for general video understanding, we observe that explicitly forcing the model to reason before answering leads to a degradation in performance. This is likely attributable to the

| Model | Reaso-ning | General Understanding Tasks | | | | Captioning Tasks | |
| | | Video-MME Overall | Video-MME S / M / L | LongVideo Bench | MV-Bench | DREAM-1K F1 / Rec / Pre | VidCapBench Acc / Pre / Cov / Con |
|---|---|---|---|---|---|---|---|
| Qwen2.5-VL-7B | ✗ | 59.4 | 69.0 / 58.8 / 51.0 | 50.6 | 58.7 | 30.9 / 28.3 / 34.0 | 12.1 / 48.5 / **81.8** / 4.5 |
| Qwen2.5-VL-7B | ✔ | 53.4 | 65.4 / 51.7 / 43.5 | 38.8 | 56.3 | 34.4 / 30.5 / **39.4** | 10.6 / 46.2 / 73.8 / 15.2 |
| Qwen2.5-VL-7B-SFT | ✗ | 54.3 | 64.8 / 50.9 / 47.7 | 43.9 | 55.6 | 29.3 / 29.0 / 29.6 | 11.9 / 48.2 / 81.2 / 6.1 |
| Video-R1 | ✔ | 58.4 | 69.0 / 58.1 / 48.4 | 53.3 | 58.3 | 31.6 / 29.4 / 34.1 | 11.7 / 48.3 / 81.2 / 7.3 |
| VideoChat-R1 | ✔ | 61.1 | 71.0 / 62.4 / 50.2 | 52.6 | 58.3 | 32.2 / 28.2 / 37.9 | 10.8 / 47.3 / 74.6 / **17.2** |
| VideoRFT | ✔ | 62.2 | 71.8 / 62.7 / 52.1 | 57.4 | 60.3 | 31.5 / 30.4 / 32.8 | 12.1 / 47.4 / 81.5 / 3.6 |
| VidBridge-R1 (Ours) | ✔ | **64.3** | **73.0 / 64.4 / 55.8** | **59.3** | **61.9** | **35.2 / 32.8 / 37.9** | **12.5 / 49.8** / 80.6 / 15.9 |

Table 1: Performance comparison on video general understanding and captioning tasks. S, M, L denote short, medium, and long, respectively. Rec, Pre, Acc, Cov, and Con abbreviate recall, precision, accuracy, coverage, and conciseness.

| Model | Reaso-ning | Reasoning Tasks | | | | | | |
| | | MMVU | NExT-QA | IntentQA | Causal-VidQA | Video-Holmes | DarkEvent-Infer-Test | MixVidQA-Test |
|---|---|---|---|---|---|---|---|---|
| Qwen2.5-VL-7B | ✗ | 41.9 | 77.0 | 91.3 | 63.7 | 38.0 | 73.0 | 29.0 |
| Qwen2.5-VL-7B | ✔ | 49.0 | 77.3 | 89.9 | 68.3 | 35.3 | 70.0 | 20.0 |
| Qwen2.5-VL-7B-SFT | ✗ | 52.3 | 71.4 | 84.2 | 67.1 | 36.6 | 80.0 | 33.0 |
| Video-R1 | ✔ | 50.9 | 79.8 | 90.9 | 60.9 | 36.5 | 61.0 | 27.0 |
| VideoChat-R1 | ✔ | 51.1 | 79.6 | 93.6 | 68.8 | 33.0 | 70.0 | 15.0 |
| VideoRFT | ✔ | 52.4 | 80.5 | 94.9 | 69.1 | 38.0 | 77.0 | 23.0 |
| VidBridge-R1 (Ours) | ✔ | **54.7** | **81.6** | **97.1** | **70.7** | **40.0** | **117.0** | **49.0** |

Table 2: Performance comparison on video reasoning tasks.

relative simplicity of these tasks, where imposing deliberate reasoning on a vanilla model may distort its initial correct understanding. However, after training with meticulously curated tasks, VidBridge-R1 demonstrates remarkable capabilities, achieving an overall score of 64.3 on Video-MME, with optimal performance across different video lengths. Furthermore, VidBridge-R1 outperforms the strongest baselines on both LongVideoBench and MVBench by 2.1% and 1.6%, respectively, demonstrating its strong general video understanding capabilities.

Regarding video captioning tasks, we find that forcing the model to reason before responding improves performance on DREAM-1K but degrades it on VidCapBench. This discrepancy arises because when compelled to reason, the model prioritizes output accuracy, leading to shorter captions that align more closely with the ground-truth captions in DREAM-1K, and achieving a very high precision score. In contrast, VidCapBench evaluates caption quality by using a judge model to answer questions based on model-generated textual captions, where longer captions tend to yield better results. Nevertheless, VidBridge-R1 strikes a favorable balance between the two benchmarks, ensuring both accuracy and comprehensiveness in its generated video captions.

In video reasoning tasks, VidBridge-R1 also demonstrates outstanding performance, surpassing the strongest baseline by an average of 1.9% across five conventional QA tasks. Furthermore, on our held-out testset, which follows the same task format as our designed proxy tasks but employs entirely distinct and unseen data, with evaluation metrics consistent with Equations 3 and 4, VidBridge-R1 achieves significant improvements. This indicates that the model has already developed a dual capability for both the divergent holistic comprehension of video content and the convergent pinpointing of key information, thereby supporting effective performance in both captioning and QA tasks within general and reasoning scenarios.

## 6.3 Ablation Study on the Training Tasks

In Table 3, we perform an in-depth analysis of how different training tasks influence model performance. When trained exclusively on the captioning task, the model achieves substantial gains on the DREAM-1K benchmark, but performs poorly on other tasks. Conversely, training solely on the VideoQA task leads to improvements on conventional QA benchmarks, yet the model still underperforms on the captioning task.

| Task Composition | | | | General Understanding Tasks | | | Reasoning Tasks | | | | Caption Tasks |
|---|---|---|---|---|---|---|---|---|---|---|---|
| Video-QA | DarkEvent-Infer | MixVid-QA | Caption | Video-MME | LongVideo-Bench | MV-Bench | MMVU | Intent-QA | DarkEvent-Infer-Test | MixVid-QA-Test | DREAM-1K F1 / Rec / Pre |
| – | – | – | ✔ | 58.0 | 41.9 | 53.5 | 50.6 | 92.5 | 64.0 | 16.0 | 34.8 / 30.6 / **40.4** |
| ✔ | – | – | – | 63.2 | 56.4 | 58.7 | 53.8 | 96.4 | 60.0 | 24.0 | 31.7 / 29.5 / 34.3 |
| ✔ | – | – | ✔ | 54.8 | 54.7 | 57.2 | 52.5 | 96.4 | 69.0 | 13.0 | 30.6 / 28.1 / 33.7 |
| ✔+ | – | – | ✔+ | 61.5 | 56.4 | 59.6 | 53.2 | 96.5 | 46.0 | 16.0 | 33.0 / 28.9 / 38.5 |
| ✔ | ✔ | – | – | 63.4 | 57.4 | 59.6 | 53.3 | 97.0 | 113.0 | 41.0 | 34.7 / 31.1 / 39.3 |
| ✔ | – | ✔ | – | 63.5 | 57.9 | 60.2 | 53.1 | 97.0 | 107.0 | 51.0 | 32.3 / 28.2 / 37.9 |
| ✔ | ✔ | ✔ | – | 63.8 | 58.6 | 60.4 | 54.1 | **97.2** | **121.0** | **54.0** | 32.2 / 28.3 / 37.4 |
| – | ✔ | ✔ | ✔ | 60.7 | 51.4 | 56.1 | 51.7 | 81.6 | 117.0 | 52.0 | 34.9 / 32.4 / 37.8 |
| ✔ | ✔ | ✔ | ✔ | **64.3** | **59.3** | **61.9** | **54.7** | 97.1 | 117.0 | 49.0 | **35.2** / **32.8** / 37.9 |

Table 3: Ablation study on the training task composition. ✔+ indicates that we expanded the volume of VideoQA and Caption task in their original proportions to match the total data volume used in the final training setup with four tasks, thereby ablating the effect of data volume.

However, naively mixing these two tasks, while seemingly reasonable, results in performance degradation across both tasks, even with increased data volume (line 4). We attribute this phenomenon to a conflict in the learning paradigms between QA and captioning under RL, an indirect optimization method guided by global reward signals. Specifically, QA is a convergent reasoning task that requires the model to output a unique, low-entropy correct answer from abundant contextual information. In contrast, captioning is a divergent generation task that demands the production of diverse and comprehensive high-entropy descriptions. The opposing task objectives may lead to inherent conflicts during RL optimization.

By progressively incorporating our proposed intermediate proxy tasks, DarkEventinfer and MixVidQA, which are designed to mitigate these issues, the model exhibits substantial performance improvements across all evaluated tasks. Upon further incorporating the captioning task, although slight performance degradation is observed on some reasoning tasks, a remarkable improvement is achieved in captioning. Collectively, these findings demonstrate that VidBridge-R1, through the strategic integration of intermediate proxy tasks, effectively bridges various video understanding tasks, demonstrating exceptional comprehensive capabilities and broad adaptability.

## 6.4 TRAINING DYNAMICS

As illustrated in Figure 3, VidBridge-R1 exhibits distinct training dynamics across different task categories, reflecting the varying cognitive demands of each task. For QA tasks in general video understanding (Video-MME, LongVideoBench), the model demonstrates rapid performance improvement during the initial training phase, which can be attributed to the relatively simpler nature of these tasks, facilitating easier learning. For video reasoning tasks (NExT-QA, MMVU, Causal-VidQA, DarkEventInfer, MixVidQA), the model

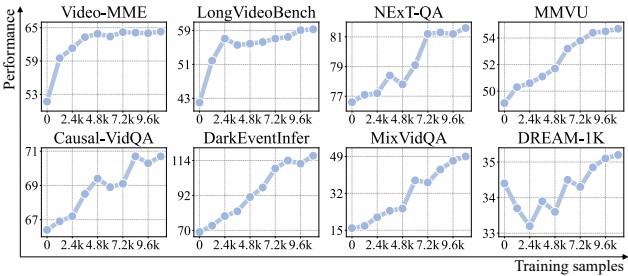

Figure 3: The training dynamics of VidBridge-R1 on video general understanding, reasoning, and captioning tasks.

shows steady and progressive performance enhancement, validating that our training framework can consistently activate the model's reasoning capabilities. In the video captioning task (DREAM-1K), the model's performance initially declines before exhibiting oscillatory improvement, suggesting that the integration of reasoning capabilities into caption generation is more intricate compared to QA tasks. Despite these varied training trajectories across tasks, VidBridge-R1 ultimately achieves sustained and robust performance gains across all tasks, underscoring the effectiveness of our meticulously constructed intermediate proxy tasks in facilitating comprehensive video understanding.

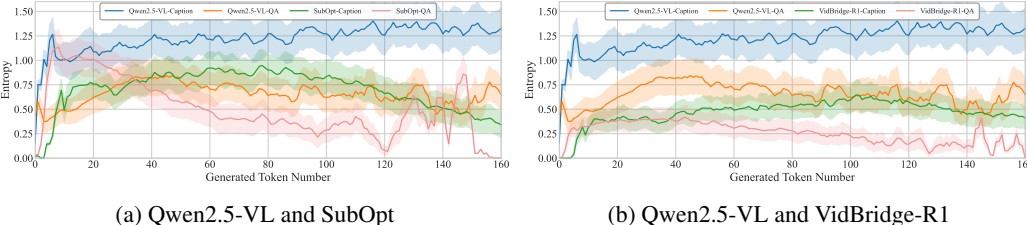

(a) Qwen2.5-VL and SubOpt        (b) Qwen2.5-VL and VidBridge-R1

Figure 4: Distribution of output token entropy across different models on captioning and QA tasks. SubOpt denotes the suboptimal model trained exclusively on conventional QA and captioning tasks. For finer detail, please zoom in.

## 6.5 Analysis of the Output Entropy

To more intuitively illustrate the inherent conflict between the Caption and QA tasks during optimization, we present the distribution of output token entropy for both tasks across different models in Figure 4. To mitigate the confounding effects of varying video inputs, all experiments are conducted on the Video-MME benchmark. For the QA task, we use the original QA pairs provided in the benchmark; for the captioning task, we prompt each model to generate descriptive captions for the same set of videos.

The results reveal that, for the original Qwen2.5-VL, a substantial gap exists between the output entropies on the Caption and QA tasks, making simultaneous optimization of these two tasks challenging. As shown in Figure 4a, when trained solely with conventional VideoQA and Caption tasks, the resulting suboptimal model still exhibits a significant entropy disparity between the two tasks during the critical generation phase (tokens #60 to #120). Moreover, during the initial generation stage, the entropy of the QA task is adversely influenced by the Caption task, rising to a abnormal level comparable to that of the base model on the Caption task. In contrast, when our proposed proxy tasks are introduced, as depicted in Figure 4b, the entropy gap between the Caption and QA tasks is markedly narrowed, thereby enabling more effective joint optimization without compromising task-specific performance.

## 7 Conclusion

In this work, we identify the challenge of the paradigm conflict between the convergent QA tasks and the divergent captioning tasks during RL for video understanding models. We show that naively combining their reward signals leads to performance degradation in both tasks. To mitigate this issue, we introduce two novel intermediate proxy tasks, DarkEventInfer and MixVidQA, designed to bridge the gap by encouraging the model to simultaneously develop holistic contextual understanding and precise information localization capabilities. Building upon this approach, we introduce VidBridge-R1, the first versatile video reasoning model that reconciles the divergence between QA and captioning tasks during RL training. Extensive experiments demonstrate that VidBridge-R1 effectively alleviates the task conflict and achieves significant performance improvements across diverse general video understanding, reasoning, and captioning benchmarks, highlighting its strong multi-task and generalization ability.

## Acknowledgements

This work is supported by the National Natural Science Foundation of China (92570204, 62576339, 62236010).

## Reproducibility Statement

The majority of training and inference details are described in Section 6.1.3. The construction process of the training data is presented in Section 4 and Appendix B.3. In addition, the prompts used for generating training data and for LLM-based evaluation are provided in Appendix B.2.

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

## A  THE USE OF LLMS

Throughout the coding and debugging stages, we leveraged LLMs for technical guidance. Following the collaborative drafting of the manuscript, we again engaged LLMs to polish and refine its language and overall expression.

## B  ADDITIONAL EXPERIMENTAL DETAILS

### B.1  DETAILS OF BENCHMARKS

To comprehensively evaluate the model performance, we selected representative benchmarks to assess the model's capabilities in general video understanding, video reasoning, and video captioning, respectively.

For general video understanding, we conduct experiments on Video-MME (Fu et al., 2024), LongVideoBench (Wu et al., 2024) and MVBench (Li et al., 2024a).

- **Video-MME** is a comprehensive benchmark for evaluating MLLMs across diverse video types and temporal lengths. It features 900 manually annotated videos spanning 254 hours and 2,700 QA pairs, offering a rigorous test of models' general understanding ability. We evaluate Video-MME without subtitles in our experiments.

- **LongVideoBench** is designed to evaluate the long-form multimodal perception and relation capability of MLLMs. It includes 3,763 web-collected videos spanning various lengths and themes and 6,678 human-annotated multiple-choice questions, distributed across 17 fine-grained categories, which assess different aspects of video-language understanding.

- **MVBench** is designed to evaluate the temporal understanding capabilities of MLLMs through 20 challenging video tasks that go beyond static image reasoning. By systematically transforming static tasks into dynamic ones, it covers a wide range of temporal skills and ensures fair evaluation using ground-truth annotations converted into multiple-choice questions.

For video reasoning tasks, we conduct experiments on MMVU (Zhao et al., 2025), NExT-QA (Xiao et al., 2021), IntentQA (Li et al., 2023), Causal-VidQA (Zang et al., 2023), Video-Holmes (Cheng et al., 2025), and our held-out test set DarkEventInfer-Test and MixVidQA-Test.

- **MMVU** is designed to evaluate the expert-level video reasoning ability of MLLMs. It contains 3,000 expert-annotated questions over 1,529 videos, which span 27 subjects from four core disciplines: Science, Healthcare, Humanities & Social Sciences, and Engineering.

- **NExT-QA** tests the model's reasoning ability over causal, temporal, and descriptive question types. In our experiments, we used the validation split containing 4,996 video-question pairs with five answer options.

- **IntentQA** contains 4,303 videos and 16k multiple-choice QA pairs focused on reasoning about people's intent in the video. We perform a zero-shot evaluation on the test set containing 2k questions.

- **Causal-VidQA** requires the model to answer questions including scene description, evidence reasoning, and commonsense reasoning. Moreover, for the commonsense reasoning questions, the model is required to not only provide a right answer but also offer a proper reason justifying why that answer is true. We present the average results across all categories.

- **Video-Holmes** evaluates complex reasoning capabilities through 1,837 QA pairs requiring strong reasoning skills, with a focus on suspenseful short films.

- **DarkEventInfer-Test and MixVidQA-Test** are test sets randomly sampled from our constructed video reasoning dataset, each containing 100 samples. They are designed to evaluate the model's capability of performing contextual reasoning across video sequences and distinguishing between different video sources when answering questions, respectively.

For video captioning tasks, we conduct experiments on DREAM-1K (Wang et al., 2024a) and Vid-CapBench (Chen et al., 2025f).

---

**Prompts for eliciting model reasoning**

{original prompts} + "Output the thinking process in `<think>` `</think>` and final answer in `<answer>` `</answer>` tags, i.e., `<think>` reasoning process here `</think>` `<answer>` answer here `</answer>`."

---

Figure 5: Prompts for eliciting model reasoning

---

**Prompts to generate QA pairs in MixVidQA**

Please create ten question-answer (QA) pairs that require reasoning across multiple segments of the video content. Avoid using terms like "frames" in your QA generation.

Guidelines:
1. **Provide clear and specific references in each QA pair.** In your generated questions, AVOID ambiguous pronouns like "the video", "the background", "the scene", "the person", "the environment", "the man", "the woman", etc. Each QA pair should stand independently with explicit references.
2. The questions must integrate information from multiple segments of the video.
3. Ensure the questions involve complex reasoning, such as deducing outcomes based on the events or details presented in specific segments.
4. Avoid creating questions that can be answered without watching the video.
5. Ensure each question has a clear, unambiguous, and definitive answer. Do not generate questions with uncertain or speculative answers.
6. Exclude subjective questions or those involving keywords such as emotional, spiritual, contribution, importance, or implication.
7. Do not generate true/false or yes/no questions.
8. Format the QA pairs in the following JSON structure: [{"question": "xxx", "answer": "xxx"}, ...]

---

Figure 6: Prompts to generate QA pairs in MixVidQA.

- **DREAM-1K** is a challenging benchmark for detailed video description, featuring 1,000 clips from diverse sources such as films, stock footage, and short-form videos. Each video is paired with fine-grained human-annotated descriptions, and evaluated using AutoDQ, a metric better suited for assessing rich, multi-event narratives than traditional captioning scores.

- **VidCapBench** comprises 643 videos designed to evaluate video captions from the perspective of their utility for text-to-video generation. Unlike DREAM-1K, which provides ground truth captions, VidCapBench assesses caption quality by inputting both the captions and carefully designed evaluation questions into a judge model. The quality of the captions is then determined by the accuracy of the judge model's responses to these questions.

### B.2 DETAILS OF PROMPTS

#### B.2.1 PROMPTS TO ELICIT REASONING

In Figure 5, we present the prompts that we used to elicit the model reasoning before response.

#### B.2.2 PROMPTS TO GENERATE QA PAIRS IN MIXVIDQA

In Figure 6, we present the prompts that we used to generate the initial QA pairs for MixVidQA.

#### B.2.3 PROMPTS TO EVALUATE DARKEVENTINFER

In Figure 7, we present the prompts that we used to evaluate model response in DarkEventInfer.

---

**Prompts to evaluate model response in DarkEventInfer**

You are an evaluator tasked with determining if a given response matches the Ground Truth (GT) provided. Your job is to compare the response and GT carefully and return a value based on their consistency.

Instructions:
1. Read the Response and GT carefully: Ensure you understand both the response and the GT completely.
2. Evaluate the Consistency:
*Score 2: If the response semantically covers the GT entirely, even if the response is longer.
*Score 1: If the response partially covers the GT, but does not fully encompass it.
*Score 0: If the response is entirely different and irrelevant from the GT or the response is None.

Response: {model response}
GT: {ground truth}

Your judgment:

---

Figure 7: Prompts to evaluate model response in DarkEventInfer.

---

**Prompts to evaluate model response in MixVidQA**

Given a question along with its ground truth and a generated answer, please judge whether the generated answer is True or False. If the ground truth or the generated answer is ambiguous, consider it as False.

Question: {question}
Groung truth: {ground truth}
Generated answer: {model response}

Your judgment:

---

Figure 8: Prompts to evaluate model response in MixVidQA.

### B.2.4 PROMPTS TO EVALUATE MIXVIDQA

In Figure 8, we present the prompts to evaluate model response in MixVidQA.

### B.3 DETAILS OF THE DATA CURATION

This subsection delineates the construction pipelines for DarkEventInfer and MixVidQA, as illustrated in Figure 9.

DarkEventInfer presents the model with a video containing black-screen segments and requires it to infer the events that occur during the masked periods based on contextual cues. Based on the event captions and their corresponding timestamp annotations from COIN (Tang et al., 2019), we randomly select one event per video (e.g., Event 4 in Figure 9) and add a black-screen mask to it. The model is then asked to reason about and predict the masked event based on the contextual information from the unmasked segments.

The MixVidQA task is designed to present the model with interleaved video clips from two distinct sources and require it to answer questions based on the most relevant video source while ignoring the other. Specifically, we source videos from Kinetics (Kay et al., 2017), each lasting around 10 seconds, which are rich in actions. To construct the mixed sequences, two videos are randomly

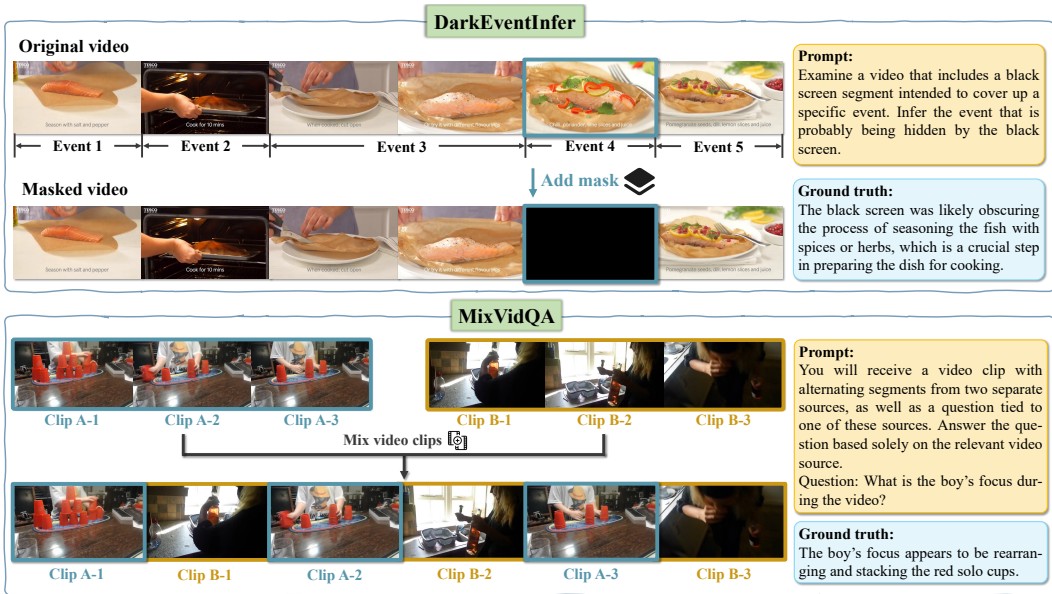

Figure 9: Details of the data curation of DarkEventInfer and MixVidQA.

selected and divided into clips with durations ranging from 1.5 to 2 seconds, which are then inter-leaved to form a new composite video. For each mixed video, we generate a set of QA pairs using Qwen2-VL-72B (Wang et al., 2024b), where each question explicitly refers to one of the original videos. The model is expected to identify the relevant video clips to provide the correct answer based on the mixed video sequences.

For the above data, we have conducted manual verification of all annotations. During this process, we corrected inaccuracies or ambiguities in the automatically generated annotations and removed instances that were excessively challenging or ambiguous even for human annotators. This ensures that the final datasets maintain high quality and are suitable for effective model training.

### B.4 DETAILS OF THE KEYWORDS REWARD

In Equation 6, we define the keyword reward by introducing the time-series-related keyword set $T$ and the speculation-related keyword set $S$, which guide the model to generate captions that are both temporally coherent and free from subjective speculation. In Figure 10, we illustrate examples of the keywords in these two sets specifically.

---

**Examples of the keywords set in keywords reward**

**Temporally-relevant set $T$:**
{"start with", "then", "next", "after", "begin with", "followed by", "following", "subse-quently", "initially", "first", "second", "finally", "lastly", ...}

**Speculation-related set $S$:**
{"possibly", "likely", "appears to", "seems to", "might", "may", "potentially", "probably", "implying", "perhaps", "presumably", ...}

---

Figure 10: Illustrative examples of the keyword sets utilized in the keywords reward.

# C ADDITIONAL EXPERIMENTAL ANALYSIS

## C.1 ABLATION ON THE JUDGE MODEL FOR CAPTIONING TASK

In the main text, we employ GPT-3.5-Turbo as the judge model for the captioning task. To evaluate the impact of different judge models on the training process, we present the experimental results obtained when using GPT-4.1[2] as the judge model in Table 4.

| Judge Model | DREAM-1K F1 / Rec / Pre | VidCapBench Acc / Pre / Cov / Con |
|---|---|---|
| GPT-3.5-Turbo | 34.8 / 30.6 / 40.4 | 13.0 / 47.8 / 80.8 / 13.6 |
| GPT-4.1 | 34.9 / 32.4 / 37.8 | 13.3 / 49.5 / 79.2 / 8.7 |

Table 4: Ablation on the judge model for the captioning task.

The experimental results indicate that the performance of using GPT-3.5-Turbo and GPT-4.1 as judge models is quite close. This suggests that the performance improvement achieved by our method primarily stems from the effectiveness of our training strategy and the high-quality training data. As long as the judge model possesses sufficient capability for evaluation, the experimental results are not critically sensitive to the specific choice of the judge model.

We opt to use OpenAI's API service for judgement in the captioning task, rather than deploying a local Qwen model as done in other tasks, because the AutoDQ (Wang et al., 2024a) evaluation method involves frequent calls to the judge model. If implemented locally, this would significantly reduce training efficiency and lead to an unacceptable increase in training time. In comparison, utilizing OpenAI's API greatly enhances training efficiency. Therefore, considering both computational efficiency and economic cost, we ultimately select GPT-3.5-Turbo as the judge model for the captioning task.

## C.2 ANALYSIS OF THE CAPTION REWARD

In the design of the caption reward, several crucial strategies have been implemented to avert undesirable behaviors and enhance the overall quality of generated captions. To prevent the model from hacking the precision reward by generating overly brief captions, we set the coefficient $\alpha$ of the precision term in Equation 5

| Coefficient $\alpha$ and $\beta$ of the Caption Reward | DREAM-1K F1 / Rec / Pre | VidCapBench Acc / Pre / Cov / Con |
|---|---|---|
| $\alpha = 1.0, \beta = 0$ (reward hacking) | 37.0 / 29.9 / 48.6 | 10.3 / 44.7 / 76.2 / 12.3 |
| $\alpha = 0.5, \beta = 0$ (w/o keywords) | 32.9 / 30.6 / 35.6 | 12.2 / 47.1 / 80.4 / 12.2 |
| $\alpha = 0.5, \beta = 0.2$ (Ours) | 34.8 / 30.6 / 40.4 | 13.0 / 47.8 / 80.8 / 13.6 |

Table 5: Analysis on the caption reward.

to 0.5. Additionally, to promote temporally coherent caption generation and suppress speculative or irrelevant content unrelated to the video, we incorporate the keywords reward as defined in Equation 6. To comprehensively validate the effectiveness of our reward design, we conduct ablation studies by training models solely on the captioning data, with their respective performances summarized in Table 5.

When $\alpha$ is set to 1, equal to the weight of the recall term, the model achieves a notable increase in precision by producing extremely concise captions. However, this improvement is largely superficial, as it comes at the cost of reduced descriptive richness and contextual coverage. Moreover, the model performs significantly worse on the QA-based captioning benchmark VidCapBench, indicating a degradation in overall caption quality. Therefore, it is essential to reduce the weight of the precision reward to avoid reward hacking.

In the absence of the keywords reward, the model's performance deteriorates compared to when the reward is included. This indicates that the keywords reward plays a pivotal role in guiding the model to generate temporally structured and semantically grounded captions, thereby substantially improving performance on the video captioning task.

---

[2]https://openai.com/index/gpt-4-1

| Coefficient $\gamma$ of KL Divergence | General Understanding Tasks | | | Reasoning Tasks | | | | Caption Tasks |
|---|---|---|---|---|---|---|---|---|
| | Video-MME | LongVideo-Bench | MVBench | NExT-QA | IntentQA | DarkEvent-Infer-Test | MixVid-QA-Test | DREAM-1K F1 / Rec / Pre |
| 0 | **64.3** | **59.3** | **61.9** | **81.6** | **97.1** | **117.0** | **49.0** | **35.2** / **32.8** / 37.9 |
| 0.05 | 63.5$_{(-0.8)}$ | 56.2$_{(-3.1)}$ | 57.9$_{(-4.0)}$ | 81.3$_{(-0.3)}$ | 96.0$_{(-1.1)}$ | 91.0$_{(-26)}$ | 39.0$_{(-10)}$ | 34.9 / 30.7 / **40.5** |
| 0.10 | 62.6$_{(-1.7)}$ | 54.1$_{(-5.2)}$ | 55.0$_{(-6.9)}$ | 80.8$_{(-0.8)}$ | 95.7$_{(-1.4)}$ | 85.0$_{(-32)}$ | 24.0$_{(-25)}$ | 34.7 / 30.6 / 40.1 |

Table 6: Ablation study on the coefficient of KL divergence.

| Format Reward | General Understanding Tasks | | | Reasoning Tasks | | | | Caption Tasks |
|---|---|---|---|---|---|---|---|---|
| | Video-MME | LongVideo-Bench | MVBench | NExT-QA | IntentQA | DarkEvent-Infer-Test | MixVid-QA-Test | DREAM-1K F1 / Rec / Pre |
| None | 64.1 | 57.8 | 58.2 | 81.2 | 96.2 | 110.0 | 47.0 | 33.1 / 31.3 / 35.0 |
| Explicit | 63.5 | 58.2 | 59.4 | 80.3 | 96.7 | 112.0 | **49.0** | 33.8 / 31.2 / 36.8 |
| Implicit | **64.3** | **59.3** | **61.9** | **81.6** | **97.1** | **117.0** | **49.0** | **35.2 / 32.8 / 37.9** |

Table 7: Ablation study on the format reward.

## C.3 ABLATION STUDY ON THE KL DIVERGENCE

In the objective function of the GRPO algorithm (Equation 2), the KL divergence constraint term is introduced to limit the deviation between the current policy model and the previous policy model, thereby ensuring the stability of the training process. However, during our training, we observed that discarding the KL divergence constraint would be a preferable option when aiming to more rapidly stimulate the model's reasoning capabilities. Table 6 presents our ablation experiments on the coefficient of the KL divergence constraint, demonstrating that stronger constraints lead to slower model convergence, ultimately impairing final performance. Consequently, when using a fixed number of training samples, omitting the KL divergence constraint better facilitates the model's reasoning abilities and yields optimal performance.

## C.4 ABLATION STUDY ON THE FORMAT REWARD

During training, we adopt an implicit format reward, as defined in Equation 9. This mechanism ensures that the models are eligible to receive any of the task-specific rewards only when its output adheres to the required format. To evaluate the effectiveness of this approach, we compare two alternatives in Table 7: one without format reward (denoted as "None"), and another with explicit format reward, where the format reward is incorporated as a separate term into the overall reward function alongside the task-specific reward.

Experimental results indicate that explicitly incorporating the format reward leads to a slight performance improvement compared to the setting without format reward in most scenarios. However, both approaches are outperformed by the implicit format reward mechanism. The underlying reason is that the implicit format reward guides the model to gradually learn the correct output format while generating accurate answers, thereby preventing the model from converging to suboptimal solutions that either satisfy only the format constraints or focus solely on answer correctness, promoting more balanced and effective training.

## C.5 ANALYSIS OF VIDBRIDGE-R1'S REASONING PATTERNS

Figure 11 presents illustrative examples of VidBridge-R1's reasoning patterns across both VideoQA and captioning tasks. In the VideoQA task, VidBridge-R1 effectively integrates the analysis of the video content with logical inference throughout its reasoning process, ultimately arriving at the correct answer. In the captioning task, VidBridge-R1 initially provides fragmented observations of the characters and events depicted in the video during the reasoning process. These partial insights are then synthesized into a comprehensive and coherent video caption in the final answer. It is noteworthy that these reasoning patterns emerge naturally from the model itself, rather than being forcibly imitated through SFT, further demonstrating the effectiveness of our training framework.

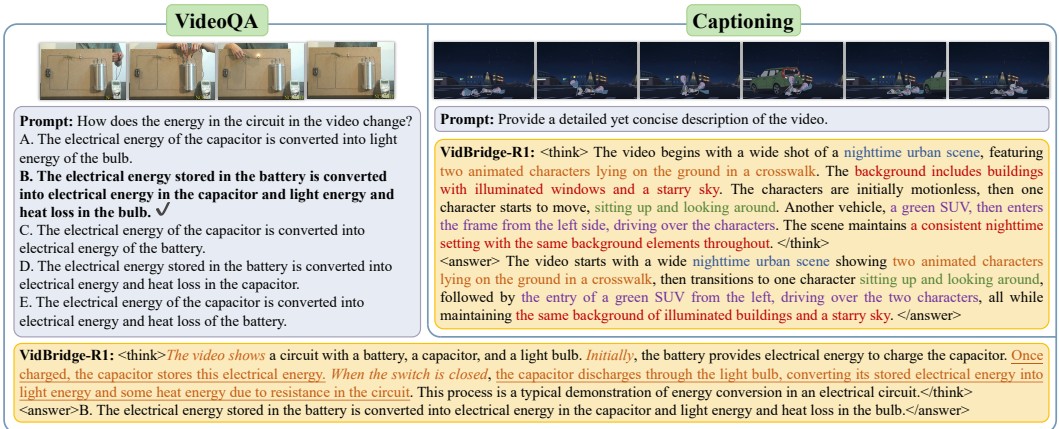

Figure 11: Illustrative examples showcasing the reasoning patterns of VidBridge-R1. In VideoQA tasks, the *keywords* indicating video content descriptions appear in italics, while reasoning steps are underlined. In captioning tasks, corresponding elements in the reasoning process and the final answer are marked with the same color.

## D    FUTURE WORK

Hallucination remains a critical challenge that substantially undermines the credibility of the model (Liu et al., 2025a; Chen et al., 2025e; Li et al., 2025a). Future work will explore effective strategies for detecting and mitigating hallucinations in both the chain-of-thought reasoning process and the final outputs generated by VidBridge-R1, with the goal of further improving the model's reliability and trustworthiness.

## E    ATTEMPTS OF SFT COLD-START BEFORE RL

Some recent studies (Feng et al., 2025; Zhang et al., 2025b; Peng et al., 2025a; Yang et al., 2025) suggest that employing SFT as a cold-start strategy to teach the model reasoning pattern before RL, can effectively improve its reasoning capabilities. In this section, we aim to evaluate the effectiveness of this approach within our proxy task framework by constructing reasoning chains for the tasks presented in the main text and conducting experiments in which the model is first initialized through SFT, followed by RL-based training.

### E.1    CURATION OF THE REASONING CHAINS

To avoid data overlap between the SFT and RL phases, we utilize the data filtered out in the main text due to high similarity among model responses for the SFT phase, while maintaining consistency with the data used in the RL phase as per the main text. In the following, we will detail the construction methods of the reasoning chains for SFT across all tasks, with the workflow illustrated in Figure 12.

#### E.1.1    CONSTRUCTION OF REASONING CHAINS FOR QA TASKS

For the QA tasks, including DarkEventInfer, MixVidQA, and conventional VideoQA, we employ the following methodology to construct the reasoning chains used in the SFT phase.

First, we feed each video along with its corresponding query and ground truth into the Qwen2-VL-72B. The model is instructed to generate a video caption tailored to the query and ground truth, accompanied by a reasoning process and a final conclusion. The generated captions, reasoning steps, and conclusions are then fed into Qwen2-72B, which filters out samples where the reasoning or conclusion is logically inconsistent with the caption, ensuring that only those captions containing sufficient and relevant information to support the reasoning and conclusion are retained.

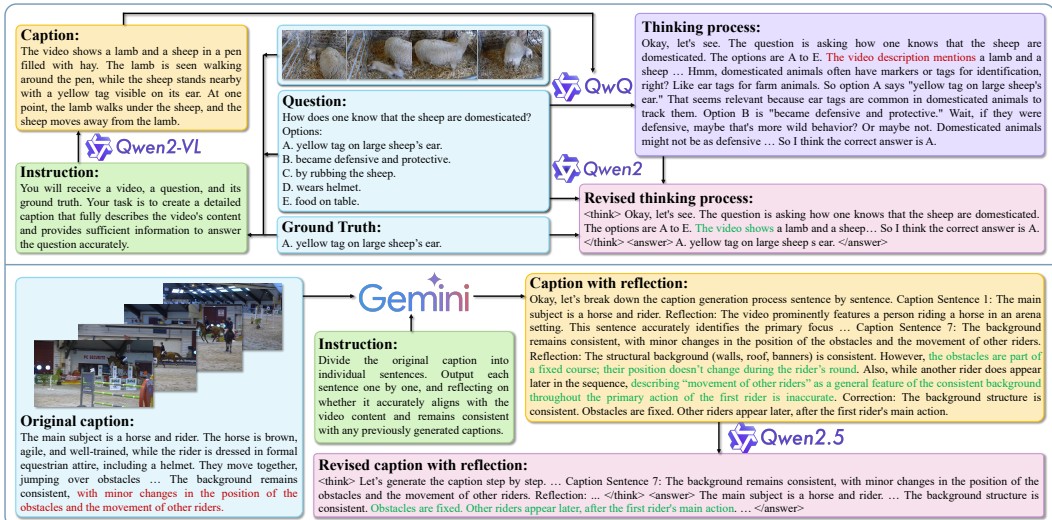

Figure 12: Pipeline for constructing reasoning chains. The upper part illustrates the process for the QA task, and the lower part shows the corresponding procedure for the caption task. For clarity, the data filtering step for non-compliant entries is omitted.

Based on the filtered data, we prompt QwQ[3] to generate more natural, detailed, and structurally clear reasoning processes using the captions and queries as input. Finally, the query, ground truth, and the reasoning chains produced by QwQ are jointly fed into Qwen2-72B, which further refines any erroneous reasoning steps that are not corrected in subsequent reasoning steps, while preserving the original structure as much as possible. During this stage, any occurrences of the phrase such as "video caption" are replaced with the term "video" to better support SFT for multimodal large language models.

To ensure data quality, we manually review 20% of the curated samples, confirming both logical consistency and linguistic accuracy to meet the training requirements.

### E.1.2 CONSTRUCTION OF REASONING CHAINS FOR THE CAPTIONING TASK

For captioning tasks, the pipeline used to construct reasoning chains used in QA tasks is no longer applicable. Instead, we leverage the strong capabilities of Gemini-2.5-Pro to construct high-quality reasoning chains.

Specifically, we input both the video and its original caption into Gemini, which is prompted to decompose the caption into individual sentences and generate them sequentially. After each generated sentence, Gemini is instructed to reflect on whether the current output conflicts with the video content or with previously generated sentences. If a conflict is detected, the model is required to immediately correct the inconsistency, disregarding the original caption, and continuing to generate new captions along with corresponding reflections. If no conflict is identified, the model should proceed to the next sentence from the original caption and perform another reflection step.

To further enhance the linguistic fluency of the reasoning process, we feed the generated caption and reflection sequences into Qwen2.5-72B. This model is instructed to preserve the logical structure of the original reasoning while rephrasing and polishing the language, thereby aligning the overall process more closely with the natural cognitive mechanism of "generating while reflecting" that characterizes human caption creation.

| Phase | DarkEventInfer | MixVidQA | VideoQA | Captioning | Total |
|---|---|---|---|---|---|
| SFT | 1,576 | 1,203 | 4,639 | 10,891 | 18,309 |
| RL | 1,841 | 2,332 | 2,003 | 4,624 | 10,800 |

Table 8: Data distribution in our training set.

---

[3] https://qwenlm.github.io/blog/qwq-32b-preview

| Model | Reaso-ning | General Understanding Tasks | | | Reasoning Tasks | | | | Caption Tasks |
|---|---|---|---|---|---|---|---|---|---|
| | | Video-MME | LongVideo-Bench | MVBench | NExT-QA | IntentQA | DarkEvent-Infer-Test | MixVid-QA-Test | DREAM-1K F1 / Rec / Pre |
| Qwen2.5-VL-7B | ✗ | 59.4 | 50.6 | 58.7 | 77.0 | 91.3 | 73.0 | 29.0 | 30.9 / 28.3 / 34.0 |
| Qwen2.5-VL-Ans-SFT | ✗ | 54.3 | 43.9 | 55.6 | 71.4 | 89.9 | 70.0 | 20.0 | 29.3 / 29.0 / 29.6 |
| Qwen2.5-VL-CoT-SFT | ✔ | 55.2 | 50.6 | 49.5 | 78.2 | 86.2 | 77.0 | 35.0 | 27.6 / 27.9 / 27.3 |
| Qwen2.5-VL-CoT-SFT-RL | ✔ | 57.0 | 52.2 | 50.5 | 77.8 | 88.1 | 80.0 | 39.0 | 28.4 / 29.5 / 27.3 |
| VidBridge-R1 (Ours) | ✔ | 64.3 | 59.3 | 61.9 | 81.6 | 97.1 | 117.0 | 49.0 | 35.2 / 32.8 / 37.9 |

Table 9: Performance comparison of different training methods. "Ans-SFT" refers to SFT using direct answers without reasoning chains. "CoT-SFT" denotes SFT with reasoning chains. "CoT-SFT-RL" stands for RL based on the CoT-SFT model.

As in the QA tasks, we manually review 20% of the generated samples to ensure both logical consistency and linguistic accuracy, confirming that the data meet the quality standards required for training. The final numbers of training samples used in the SFT and RL phases are summarized in Table 8.

### E.2 EXPERIMENTAL RESULTS

In Table 9, we present the experimental results related to the approach of *SFT-then-RL*. Specifically, we compare VidBridge-R1 with three additional settings: (1) SFT using only direct answers without reasoning chains (Qwen2.5-VL-Ans-SFT), (2) SFT with the inclusion of reasoning processes (Qwen2.5-VL-CoT-SFT), and (3) further applying RL after SFT with reasoning processes (Qwen2.5-VL-CoT-SFT-RL).

The results demonstrate that, across general video understanding, video reasoning, and captioning tasks, directly applying RL significantly outperforms the approach involving an initial phase of SFT. This performance gap can be attributed to the fact that, during the cold-start phase, the model is compelled to learn complex reasoning patterns that exceed its current cognitive capacity, consequently exerting a negative impact on the subsequent RL phase. Moreover, the performance improvement of RL on the base model is diminished in the scenario of CoT-SFT, further supporting our hypothesis. These findings collectively suggest that compelling the model to imitate sophisticated reasoning processes beyond its current capabilities may ultimately degrade its final performance.

### E.3 QUALITATIVE RESULTS

Figure 13 presents a qualitative comparison between two training paradigms: direct RL (VidBridge-R1) and SFT cold-start followed by RL (Qwen2.5-VL-CoT-SFT-RL). While the model initialized with SFT demonstrates reasoning patterns that appear more aligned with human cognitive processes, its outputs often contain significant hallucinations. In contrast, the model trained directly with RL tends to produce more concise and accurate reasoning content.

Specifically, in the upper case, Qwen2.5-VL-CoT-SFT-RL erroneously claims that the black screen segment lasts for an hour, despite the entire video being less than three minutes. Furthermore, it misrepresents the temporal sequence of events. In comparison, VidBridge-R1 accurately identifies the location of the black screen event and provides a correct inference. In the lower case, Qwen2.5-VL-CoT-SFT-RL hallucinates the presence of various additional ingredients besides the chicken, while VidBridge-R1 correctly recognizes that the pot contains only chicken and water.

These observations suggest that compelling the model to imitate complex reasoning processes beyond its current capability may lead to hallucinatory inferences inconsistent with the video content, thereby ultimately undermining the model's final performance.

## F ADDITIONAL CASE STUDIES

As illustrated in Figures 14, 15, and 16, we present qualitative comparisons between VidBridge-R1 and the baseline models (Video-R1, VideoChat-R1 and VideoRFT) across a range of tasks, including general video understanding, cognitive reasoning, and descriptive captioning. The results indicate that all baseline models exhibit varying degrees of hallucination during their lengthy reasoning chains, while VidBridge-R1 consistently delivers more accurate and reliable performance.

## Cases on Reasoning Tasks

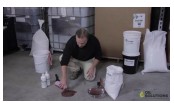 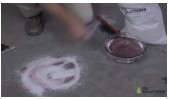 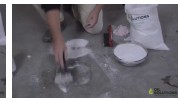 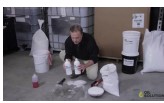  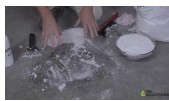

**Prompt:** You will be provided with a video featuring a black screen segment that deliberately conceals a particular event. You should analyze the video and deduce the event most likely hidden by the black screen.
**Ground Truth:** The event hidden by the black screen segment is most likely the mixing and chemical reaction between the liquid in the bottles and the materials on the ground.

**Qwen2.5-VL-CoT-SFT-RL:** <think> Okay, let's tackle this problem step by step… The key points from the video are: 1. A man is kneeling next to a white powder bag labeled "Solutions." 2. He pours the powder into a pan and spreads it around. 3. Then he adds a liquid from a bottle, mixes them, and the mixture starts reacting. 4. After mixing, there's a black screen for one hour. 5. When the screen comes back, the reaction has stopped, and the powder is gone. So the black screen is during the one-hour period after the reaction starts… So the concealed event is the ongoing reaction that uses up the powder, resulting in its complete disappearance. </think> <answer> The event hidden by the black screen segment is likely the continuation of the chemical reaction between the powder and liquid, which leads to the complete consumption of the powder. </answer>

**VidBridge-R1:** <think> The black screen segment in the video likely conceals a particular event or series of actions that occur between the visible frames. This could include the application of a solution, the reaction or setting of materials, and the cleaning and preparation steps that the person is performing. The black screen serves to hide the details of this specific process, creating a pause in the visual narrative to build anticipation for the next visible action or result. </think> <answer> The black screen segment likely hides a specific event or series of actions that occur between the visible frames, such as the mixing, application, reaction, and cleaning processes involving the substances and tools laid out on the floor. </answer>

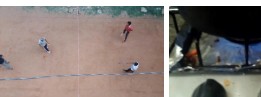 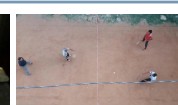 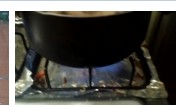 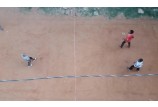 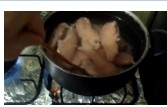

**Prompt:** You are given a video clip made up of alternating segments from two different sources, along with a question related to one of them. Your should answer the question using only the information from the relevant video source. *Question: What is the primary ingredient used alongside the meat during the cooking process?*
**Ground Truth:** Water is the main ingredient used alongside the meat for boiling.

**Qwen2.5-VL-CoT-SFT-RL:** <think> Okay, let's tackle this question. The user is asking about the primary ingredient used alongside the meat during the cooking process… so I need to identify the cooking frames… Looking at the cooking frames, it mentions that the main ingredient is chicken, along with various ingredients. So, the primary ingredient used alongside the chicken during the cooking process is another unspecified food item that complements it in the dish. Therefore, the answer to the question is that another unspecified food item is used alongside the chicken while being cooked. </think> <answer> Another unspecified food item is used alongside the chicken while being cooked. </answer>

**VidBridge-R1:** <think> The primary ingredient used alongside the meat during the cooking process is water. </think> <answer>Water is the primary ingredient used alongside the meat. </answer>

Figure 13: Qualitative comparison of direct RL (VidBridge-R1) with SFT cold-start then RL (Qwen2.5-VL-CoT-SFT-RL).

Notably, as shown in Figure 15, although Video-R1 generates a correct reasoning chain, it ultimately produces an inconsistent and incorrect conclusion. This discrepancy suggests potential limitations in the model's ability to maintain coherence over long reasoning sequences. In contrast, VidBridge-R1 achieves superior accuracy by adopting more concise reasoning chains derived from itself, rather than relying on forced imitation through SFT, which better aligns with its inherent capabilities, leading to more coherent and dependable outputs.

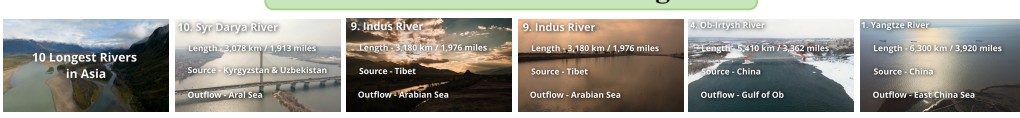

**Cases on Video Understanding Tasks**

**Prompt:** What do the scenes shot on Indus River and Yangze River have in common?
**A. They are all shot at dusk.** ✔
B. The land is covered with snow in both scene.
C. There are bridges on both scene.
D. They both contain an island in the middle of the lake.

**Qwen2.5-VL (reasoning):** <think> The Indus River scene shows a bridge, and the Yangtze River scene also shows a bridge. No other common features are shared between the two scenes. </think> <answer> C. There are bridges on both scene. </answer>

**Video-R1:** <think> Let's break down the options… Option A: "They are all shot at dusk." This option suggests that both scenes were captured during the same time of day. However, there is no explicit mention or visual evidence in the video description that indicates the time of day for either scene. Therefore, this option seems unlikely… Option C: "There are bridges on both scene." This option suggests that both scenes feature bridges. The video description clearly mentions a bridge over the Indus River and a bridge over the Yangze River. Both descriptions explicitly state the presence of a bridge, so this option appears to be correct… After evaluating all the options, the most accurate description of the scenes for the Indus River and Yangze River is that there are bridges on both scenes. Final answer: C <think> <answer> C </answer>

**VideoChat-R1:** <think> Let's analyze each option for the scenes shown: 1. Option A: They are all shot at dusk. The Indus River scene appears to be during daylight with clear skies. The Yangtze River scene also appears to be during daylight with clear skies. Neither scene shows signs of dusk… 3. Option C: There are bridges on both scenes. The Indus River scene has a bridge visible across the river. The Yangtze River scene also has a bridge visible across the river… Based on this analysis, the only commonality between the two scenes is that they both contain bridges. </think> <answer> C. There are bridges on both scene. </answer>

**VideoRFT:** <think> … The video begins with the Syr Darya River, followed by the Indus River, which is shown as a wide river with a city in the background… until reaching the Yangtze River, which is described as flowing into the East China Sea. For the Indus River, the video shows a wide river with a city in the background, indicating a bustling urban area. For the Yangtze River, the video depicts a wide river with a city in the background, also suggesting a significant urban presence. Now, let's evaluate the given options: Option A: They are all shot at dusk. The video does not specify the time of day for either the Indus or Yangtze River scenes, so this option cannot be confirmed… Option C: There are bridges on both scenes. The Indus River scene includes a bridge over the river, while the Yangtze River scene also features a bridge. This option is correct… Based on the visual evidence, the common element between the Indus and Yangtze River scenes is the presence of bridges. </think> <answer> C </answer>

**VidBridge-R1:** <think> The scenes shot on the Indus River and Yangtze River are both shown in the video, and upon analysis, both scenes are captured during dusk, as indicated by the lighting and sky conditions in each respective frame. </think> <answer> A. They are all shot at dusk. </answer>

Figure 14: Qualitative comparison of VidBridge-R1 with baseline models on QA tasks in general video understanding.

**Cases on Reasoning Tasks**

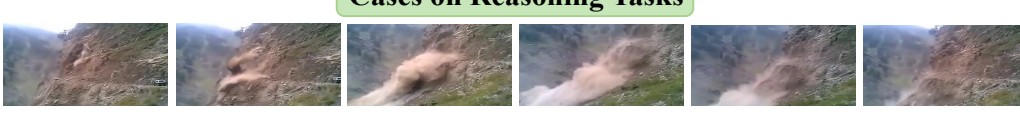

**Prompt:** Which of the following interventions would be best suited to prevent future events such as those shown in the video from occurring in the future?
A. Installing horizontal drainage systems to reduce groundwater pressure within the slope.
B. Regrading the slope to a gentler angle by removing material from the upper sections.
**C. Securing the cliff face with rock bolts and covering it with wire mesh to stabilize loose rocks.** ✔
D. Planting deep-rooted vegetation on the cliff face to reinforce the slope with root systems.
E. Diverting the road away from the cliff by constructing a tunnel through the mountain.

**Qwen2.5-VL (reasoning):** \<think\> The video depicts a landslide or a similar type of slope failure. To prevent future occurrences, the most effective intervention should address the root cause of the instability. Option A: Installing horizontal drainage systems can help manage groundwater pressure, which is often a contributing factor to slope failures… Option C: Securing the cliff face with rock bolts and wire mesh can provide immediate stability but may not address the underlying issues causing the instability… \</think\> \<answer\> A. Installing horizontal drainage systems to reduce groundwater pressure within the slope. \</answer\>

**Video-R1:** \<think\> Let's break down the options one by one to determine which intervention would be most effective for preventing future landslides or similar events… Considering the effectiveness and practicality, the best intervention would be to secure the cliff face with rock bolts and covering it with wire mesh. This provides immediate and effective stabilization. Final answer: C \</think\> \<answer\> A. Installing horizontal drainage systems to reduce groundwater pressure within the slope. \</answer\>

**VideoChat-R1:** \<think\> The video shows a landslide or a similar type of slope failure. To prevent future occurrences, the intervention should address the root cause of the instability, which could be related to water pressure, slope angle, loose rocks, or soil composition. Option A addresses groundwater pressure, which can be a significant factor in slope stability… Option C focuses on securing the slope with physical reinforcement, which is effective but may not be the most cost-effective solution… Given the immediate need to address the issue without major structural changes, installing horizontal drainage systems seems like the most direct and effective way to manage groundwater pressure and prevent future landslides. \</think\> \<answer\> A \</answer\>

**VideoRFT:** \<think\> … Option A involves installing horizontal drainage systems to reduce groundwater pressure. This could help stabilize the slope by managing water flow, which is crucial for preventing landslides caused by saturated soil. However, the video does not show any signs of water erosion, so this option might not be directly applicable… Option C proposes securing the cliff face with rock bolts and wire mesh. This method can stabilize loose rocks and prevent them from falling, but it might not address the underlying issues causing the slope instability… Given the immediate need to prevent further landslides, Option A (horizontal drainage) seems the most direct approach to manage groundwater pressure and stabilize the slope. \</think\> \<answer\> A \</answer\>

**VidBridge-R1:** \<think\>The video depicts a landslide or similar slope failure event, which is likely caused by factors such as unstable slope angles, loose rock, or lack of root reinforcement. Among the given options, the most comprehensive and effective intervention to prevent future similar events would be one that addresses multiple potential causes and provides long-term stability.\</think\> \<answer\>C. Securing the cliff face with rock bolts and covering it with wire mesh to stabilize loose rocks.\</answer\>

Figure 15: Qualitative comparison of VidBridge-R1 with baseline models on QA tasks in video reasoning.

---

**Cases on Captioning Tasks**

**Prompt:** Provide a detailed yet concise description of the video.

**Qwen2.5-VL (reasoning):** \<think\> The video depicts a scene set at night with a starry sky. Two characters, one with blue hair and another with pink hair… The overall atmosphere seems calm but slightly mysterious due to the characters' positions and the setting. \</think\> \<answer\> The video shows two animated characters in a nighttime urban setting, with one character sitting up while the other lies down. A green car passes by, partially covering the view of the characters. The scene has a calm yet mysterious ambiance. \<answer\>

**Video-R1:** \<think\> The video depicts an animated scene set at night… \</think\> \<answer\> The video begins with two animated characters lying on the ground in a nighttime urban setting. One character has blue hair and the other has pink hair… As the scene develops, the character with blue hair starts to move, sitting up and then standing. The character with pink hair remains lying down. The character with blue hair then walks away, leaving the character with pink hair still lying on the ground. A green car drives into the frame from the left side, partially obscuring the view of the characters. The video concludes with the character with pink hair still lying on the ground, while the green car continues to drive away. \<answer\>

**VideoChat-R1:** \<think\> The video depicts two animated characters lying on the ground… \</think\> \<answer\> The video shows two animated characters lying on the ground in an urban setting at night. They appear to be in distress or unconscious, as they are sprawled out on the pavement. A green car drives past them, indicating that they might have been hit by the vehicle. The background features a typical cityscape with buildings and a clear sky filled with stars, suggesting it's late at night. The overall mood of the video conveys a sense of urgency and concern for the characters' well-being. \</answer\>

**VideoRFT:** \<think\> To describe the video, I'll break it down step-by-step… \</think\> \<answer\> The video shows two animated characters lying on a a quiet, starlit urban street at night. One character has blue hair and wears a light-colored outfit, while the other has pink hair and a white and yellow outfit. Initially, they are motionless, resting near a streetlight with illuminated buildings in the background. As the video progresses, the blue-haired character begins to sit up and look around, followed by the pink-haired character who also sits up. A green vehicle with a red roof then enters the scene from behind, moving toward the characters. Throughout, the characters interact, gesturing and facing each other, while the serene, starry night and urban backdrop remain constant. \</answer\>

**VidBridge-R1:** \<think\> The video begins with a wide shot of a nighttime urban scene, featuring two animated characters lying on the ground in a crosswalk. The background includes buildings with illuminated windows and a starry sky. The characters are initially motionless, then one character starts to move, sitting up and looking around. Another vehicle, a green SUV, then enters the frame from the left side, driving over the characters who remain on the ground. The scene maintains a consistent nighttime setting with the same background elements throughout. \</think\> \<answer\> The video starts with a wide nighttime urban scene showing two animated characters lying on the ground in a crosswalk, then transitions to one character sitting up and looking around, followed by the entry of a green SUV from the left, driving over the two characters, all while maintaining the same background of illuminated buildings and a starry sky. \</answer\>

Figure 16: Qualitative comparison of VidBridge-R1 with baseline models on video captioning tasks.

