# OpenReview forum: "VidBridge-R1: Bridging QA and Captioning for RL-based Video Understanding Models with Intermediate Proxy Tasks"
_ICLR.cc/2026/Conference — ICLR 2026 Poster_

### Official Review · Reviewer_mgtE · 2025-10-15

**Soundness:** 3
**Presentation:** 3
**Contribution:** 2
**Rating:** 4
**Confidence:** 3

**Summary:**

This paper proposes two kinds of data augmentation methods for video reasoning tasks, namely DarkEventInfer and MixVidQA, enabling the model to predict the answer under the augmented video. Experiments show the effectiveness of the proposed method.

**Strengths:**

1. The idea sounds reasonable. Effective data augmentation methods are always needed.
2. The writing is clear.
3. The presented results are promising.

**Weaknesses:**

1. As an data augmentation method, whether the method is scalable is questionable.
2. Lack of detailed analysis, why the reasoning models show poor performance  on video general understanding and captioning tasks, but the model trained with the augmented reasoning data bring clear improvement. Is this a specific kind of overfitting or data leakage?

**Questions:**

Please refer to the weaknesses.

---

> ### Author Response · Authors · 2025-11-15
> **Response to the Reviewer mgtE [Concerns 1]**
>
> **Concern 1: As an data augmentation method, whether the method is scalable is questionable.**
>
> **Response 1:** Thank you for your insightful comment! Scalability indeed plays an essential role in the era of LLMs. **Our proposed proxy tasks can be easily scaled up for the following reasons.**
>
> In Sec. 4.1 and App. B.4, we describe the data construction process for the DarkEventInfer task. This task only requires event-level timestamp annotations. With such annotations, we can simply mask the corresponding video segment with a black screen and ask the model to infer the masked event based on the context. Many existing datasets already provide event-level timestamp annotations, such as TimeIT-125K [1], TimePro-349K [2], E.T. Instruct-164K [3], etc.
>
> In Sec. 4.2 and App. B.4, we illustrate the data construction process for the MixVidQA task. This task has minimal requirements on data sources, and it only needs a strong MLLM to generate referential questions for one original video, after which the two videos are interleaved at a fixed interval. The videos we use are sourced from Kinetics, which contains around 500K samples.
>
> We sincerely appreciate your valuable insight. Given that our proxy tasks are inherently scalable, we will also explore in future work whether further increasing their data volume can lead to additional performance improvements.
>
> ---
> [1] TimeChat: A Time-sensitive Multimodal Large Language Model for Long Video Understanding (CVPR 2024)
>
> [2] TimeSuite: Improving MLLMs for Long Video Understanding via Grounded Tuning (ICLR 2025)
>
> [3] E.T. Bench: Towards Open-Ended Event-Level Video-Language Understanding (NeurIPS 2025)

---

> ### Author Response · Authors · 2025-11-15
> **Response to the Reviewer mgtE [Concerns 2]**
>
> **Concern 2: Lack of detailed analysis, why the reasoning models show poor performance on video general understanding and captioning tasks, but the model trained with the augmented reasoning data bring clear improvement. Is this a specific kind of overfitting or data leakage?**
>
> **Response 2:** Thanks for your insightful question again!
>
> **In Sec. 6.5 of the revised paper, we have added an analysis of the token-entropy distributions for different models on both QA and captioning tasks.** Please refer to the updated PDF for the figures and additional discussion.
>
> For the original Qwen2.5-VL model, there exists a large gap in the output-token entropy distribution between QA and captioning tasks. **This inherent conflict makes it challenging to jointly optimize the two tasks.** If we train the model without our proxy tasks, i.e., using only standard VideoQA and captioning data, the results (Fig. 4a in the paper) show that a significant entropy gap persists in the key generation phase (#60 → #120 tokens). Moreover, at the beginning of generation, the QA-task entropy is influenced by the captioning task and even rises to the same level as the base model’s caption entropy. In contrast, when we introduce our proposed proxy tasks, as shown in Fig. 4b, **we effectively reduce the entropy gap between captioning and QA, making the joint optimization of both tasks substantially easier**.
>
> ---
>
> Meanwhile, we also conducted experiments to examine whether the observed improvements might stem from data leakage. Specifically, **to verify that our performance gains stem from the proxy tasks themselves rather than the specific datasets, we redesigned our training data by degrading the two proxy tasks back into standard VideoQA and captioning tasks**, as detailed below.
>
> For the DarkEventInfer dataset (derived from COIN), we converted it into a standard QA task by removing the black-screen mask and directly prompting the model to answer: "What happens during \<start_time\> and \<end_time\>?" We retained the same LLM-based evaluation protocol used previously.
>
> For MixVidQA, we generated high-quality captions for the original, non-interleaved videos using Gemini-2.5-Pro and treated them as standard captioning tasks for training.
>
> **In this setup, our training data retains all original training videos but excludes our designed proxy tasks entirely, enabling a clean evaluation of the proxy tasks' contribution.** The results are summarized in the table below:
>
> | Method | Video-MME | LongVideoBench | MMVU | DREAM-1K |
> |-----|-----|-----|-----|-----|
> | w/ full data, but w/o proxy tasks | 60.9 | 55.2 | 53.4 | 33.8 |
> | w/ proxy tasks (VidBridge-R1) | **64.3** | **59.3** | **54.7** | **35.2** |
>
> As shown, when training on the same data but without the proxy tasks, the model achieves significantly lower performance on both QA and captioning benchmarks compared to training with the proposed proxy tasks. **This ablation strongly confirms that the gains originate from our proxy-task design, rather than any form of data leakage.**

---

> ### Author Response · Authors · 2025-11-17
> **Friendly Reminder to Review Our Rebuttal**
>
> Dear Reviewer mgtE,
>
> We would like to kindly follow up and ask if you could take a look at our responses and reevaluate our work based on them. Please let us know whether our responses address your concerns, or if there are any additional details we can provide to help clarify. We sincerely appreciate your time and consideration!
>
> Best regards,
>
> Authors of Submission 3248

---

> ### Author Response · Authors · 2025-11-24
> **Gentle Follow-Up on Rebuttal Review**
>
> Dear Reviewer mgtE,
>
> I hope this message finds you well. We are writing to gently follow up regarding our rebuttal. We understand that this is a busy period, but if you could spare a moment to review our responses and share any updated evaluation, we would be truly grateful.
>
> Please let us know if any additional clarification or information from our side would be helpful. We sincerely appreciate your time and effort in reviewing our work.
>
> Warm regards,
>
> Authors of Submission 3248

---

> ### Author Response · Authors · 2025-11-27
> **Gentle Reminder: Discussion Period Closing Soon**
>
> Dear Reviewer mgtE,
>
> Happy Thanksgiving! I hope this message finds you well amid the holiday.
>
> As the discussion period draws to a close, we’d like to kindly remind you of our rebuttal and respectfully ask if you might have a moment to review our responses. Should you have any updated feedback or further thoughts on our submission, we would greatly appreciate it!
>
> We fully understand the heavy workload during this stage, and we remain grateful for the time and attention you have already dedicated to our submission. If any additional clarification from our side would be helpful, please don’t hesitate to let us know.
>
> Thank you once again for your time and support, and wishing you a wonderful Thanksgiving holiday!
>
> Best regards,
>
> Authors of Submission 3248

---

> ### Comment · Reviewer_mgtE · 2025-11-28
> **Response to the rebuttal**
>
> Thank the authors for the rebuttal to my concerns. The concerns of the data leakage problem is resolved and I am glad to see the proposed method can be applied to large-scale training in the future. I'd like to improve my score to 6.

---

> > ### Author Response · Authors · 2025-11-28
> > **Thanks for your acknowledgement!**
> >
> > Thank you very much for taking the time to review our rebuttal and for raising your score to 6!
> >
> > We are glad that our clarifications have adequately addressed your concerns, and we truly appreciate your positive assessment of the scalability of our method.

---

### Official Review · Reviewer_rDVT · 2025-10-20

**Soundness:** 2
**Presentation:** 3
**Contribution:** 3
**Rating:** 6
**Confidence:** 3

**Summary:**

The paper focuses on post-training a video LLM for both QA and captioning, arguing naively combining the reward signals for both tasks leads to suboptimal performance. Two proxy tasks are presented:
DarkEventInfer, which presents videos with masked event segments, forcing the model to use context to infer the missing content.
MixVidQA, which interleaves sequences from two different videos and challenges the model to reason about one while ignoring the other.
The resulting VidBridge-R1 model is trained with these proxy tasks alongside traditional QA and captioning tasks. Experiments show that VidBridge-R1 achieves sota performance across a various QA and captioning benchmarks. The paper also includes an ablation study and analysis of the training dynamics.

**Strengths:**

- Well motivated problem statement and intuitive proposals
- Sota performance on both captioning and QA tasks

**Weaknesses:**

- The ablation in Table 3 eludes important rows showing the benefit of the proposed tasks together with the caption task, as well as the row with VidMixQA and not DarkEventInfer
- Each task being based on different data makes it difficult to disentangle the benefits of the task vs the data

**Questions:**

- How much compute do the new tasks add compared to the standard ones?
- About the training dynamics study: the initial dip in captioning performance on Dream is striking. How sensitive is the final model performance to the sampling ratio between the four tasks during training?
- Is there any other evidence than the downstream performance for the conflict between the QA and captioning task?

---

> ### Author Response · Authors · 2025-11-15
> **Response to the Reviewer rDVT [Concerns 1]**
>
> **Concern 1: The ablation in Table 3 eludes important rows showing the benefit of the proposed tasks together with the caption task, as well as the row with VidMixQA and not DarkEventInfer**
>
> **Response 1:** Thank you for your valuable suggestion regarding the ablation study.
>
> We have added the two ablation experiments you mentioned, with the results shown in the table below. We have also updated Tab. 3 in the paper accordingly, and you may refer to the new PDF file for the updated version.
>
> | Video-QA | DarkEvent-Infer | MixVid-QA | Caption | | Video-MME | LongVideo-Bench | MV-Bench | MMVU | Intent-QA| DarkEvent-Infer-Test | MixVid-QA-Test | DREAM-1K |
> |-----|-----|-----|-----|-----|-----|-----|-----|-----|-----|-----|-----|-----|
> | $\checkmark$ | $\checkmark$ | **-** | **-** | | 63.4 | 57.4 | 59.6 | 53.3 | 97.0 | 113.0 | 41.0 | 34.7 |
> | $\checkmark$ | **-** | $\checkmark$ | **-** | (new) | 63.5 | 57.9 | 60.2 | 53.1 | 97.0 | 107.0 | 51.0 | 32.3 |
> | $\checkmark$ | $\checkmark$ | $\checkmark$ | **-** | | 63.8 | 58.6 | 60.4 | 54.1 | **97.2** | **121.0** | **54.0** | 32.2 |
> | **-** | $\checkmark$ | $\checkmark$ | $\checkmark$ | (new) | 60.7 | 51.4 | 56.1 | 51.7 | 81.6 | 117.0 | 52.0 | 34.9 |
> | $\checkmark$ | $\checkmark$ | $\checkmark$ | $\checkmark$ | | **64.3** | **59.3** | **61.9** | **54.7** | 97.1 | 117.0 | 49.0 | **35.2** |
>
> The results show that when training with only conventional VideoQA and MixVidQA, the model achieves slight improvements on most tasks, which is similar to the case when training with only conventional VideoQA and DarkEventInfer.
>
> When training with both proxy tasks alongside the Caption task, the model exhibits a clear performance drop on standard QA tasks. Upon examining the model outputs, we observe that, in the absence of conventional VideoQA task, the model occasionally fails to follow the multiple-choice format, resulting in evaluation failures. Meanwhile, the inclusion of the caption task improves captioning performance. **This ablation study confirms the core motivation behind our design of proxy tasks: they are not intended to directly enhance the model’s foundational capabilities on QA or captioning tasks, but rather to serve as an intermediate bridge that mitigates the optimization conflicts inherent in jointly training QA and captioning tasks, thereby reducing the difficulty of simultaneously optimizing these two distinct objectives.**

---

> ### Author Response · Authors · 2025-11-15
> **Response to the Reviewer rDVT [Concerns 2]**
>
> **Concern 2: Each task being based on different data makes it difficult to disentangle the benefits of the task vs the data**
>
> **Response 2:** Thank you very much for this thoughtful and valuable suggestion. We greatly appreciate your insight regarding the need to disentangle the contributions of the proxy tasks from those of the underlying data.
>
> To verify that our performance gains stem from the proxy tasks themselves rather than the specific datasets, **we redesigned our training data by degrading the two proxy tasks back into standard VideoQA and captioning tasks**, as detailed below.
>
> For the DarkEventInfer dataset (derived from COIN), we converted it into a standard QA task by removing the black-screen mask and directly prompting the model to answer: "What happens during \<start_time\> and \<end_time\>?" We retained the same LLM-based evaluation protocol used previously.
>
> For MixVidQA, we generated high-quality captions for the original, non-interleaved videos using Gemini-2.5-Pro and treated them as standard captioning tasks for training.
>
> **In this setup, our training data retains all original training videos but excludes our designed proxy tasks entirely, enabling a clean evaluation of the proxy tasks' contribution.** The results are summarized in the table below:
>
> | Method | Video-MME | LongVideoBench | MMVU | DREAM-1K |
> |-----|-----|-----|-----|-----|
> | w/ full data, but w/o proxy tasks | 60.9 | 55.2 | 53.4 | 33.8 |
> | w/ proxy tasks (VidBridge-R1) | **64.3** | **59.3** | **54.7** | **35.2** |
>
> As shown, when training on the same data but without the proxy tasks, the model achieves significantly lower performance on both QA and captioning benchmarks compared to training with the proposed proxy tasks. **This ablation study further confirms that our designed proxy tasks play a crucial role in simultaneously enhancing the model’s performance on both QA and captioning tasks.**

---

> ### Author Response · Authors · 2025-11-15
> **Response to the Reviewer rDVT [Concerns 3]**
>
> **Concern 3: How much compute do the new tasks add compared to the standard ones?**
>
> **Response 3:** The proposed proxy tasks introduce an additional LLM-based evaluation step compared to standard QA tasks, but each sample only requires a single LLM call. This is more resource-efficient compared to the caption evaluation pipeline, which typically extracting events and checking entailment relationships with the ground truth event list. Moreover, considering that conventional VideoQA tasks account for a smaller proportion of the training data compared to Caption task, **the inclusion of the proxy tasks actually reduces the average computational cost per training sample**.

---

> ### Author Response · Authors · 2025-11-15
> **Response to the Reviewer rDVT [Concerns 4]**
>
> **Concern 4: About the training dynamics study: the initial dip in captioning performance on Dream is striking. How sensitive is the final model performance to the sampling ratio between the four tasks during training?**
>
> **Response 4:** During training, the model’s performance on DREAM-1K initially dips and then gradually recovers and improves. This behavior is quite different from that of conventional VideoQA tasks, which we attribute to the inherent conflict between captioning and QA objectives. As the model progressively learns the proposed proxy tasks, this conflict is gradually resolved, enabling the model to rapidly improve on QA tasks while also achieving gains on captioning. This distinction also offers another perspective relevant to your Question 3.
>
> In our current training data, the ratio among DarkEventInfer : MixVidQA : conventional VideoQA : captioning is approximately 1 : 1 : 1 : 2.3. In the table below, we show results when varying the proportion of captioning data so that the ratios become 1 : 1 : 1 : 2.0 and 1 : 1 : 1 : 2.5. As demonstrated, **the final model performance is highly robust to the captioning-task proportion within this range**.
>
> | Proportion | Video-MME | LongVideoBench | MMVU | DREAM-1K |
> |-----|-----|-----|-----|-----|
> | 1 : 1 : 1 : 2.3 (VidBridge-R1) | 64.3 | 59.3 | 54.7 | 35.2 |
> | 1 : 1 : 1 : 2.0 | 63.9 | 59.1 | 54.5 | 35.3 |
> | 1 : 1 : 1 : 2.5 | 64.2 | 59.0 | 54.8 | 35.2 |

---

> ### Author Response · Authors · 2025-11-15
> **Response to the Reviewer rDVT [Concerns 5]**
>
> **Concern 5: Is there any other evidence than the downstream performance for the conflict between the QA and captioning task?**
>
> **Response 5:** Thanks for your insightful question!
>
> In Sec. 6.5 of the revised paper, we have added an analysis of the output token-entropy distributions for different models on QA and captioning tasks. **We kindly invite you to check the updated PDF for the figures and more detailed discussion.**
>
> **To briefly answer your question: beyond downstream performance, we can indeed analyze the conflict from the perspective of output token entropy.** For the original Qwen2.5-VL, there is a substantial gap between the entropy distributions of its generated tokens on QA versus captioning tasks. This intrinsic discrepancy reflects the inherent conflict and helps explain why jointly optimizing these two tasks is challenging.

---

> ### Author Response · Authors · 2025-11-17
> **Friendly Reminder to Review Our Rebuttal**
>
> Dear Reviewer rDVT,
>
> We would like to kindly follow up and ask if you could take a look at our responses and reevaluate our work based on them. Please let us know whether our responses address your concerns, or if there are any additional details we can provide to help clarify. We sincerely appreciate your time and consideration!
>
> Best regards,
>
> Authors of Submission 3248

---

> > ### Comment · Reviewer_rDVT · 2025-11-23
> > **Response to rebuttal**
> >
> > I thank the authors for their answers and confirm they resolve my concerns. I also read other reviews and responses and vote for accepting this paper.

---

> > > ### Author Response · Authors · 2025-11-23
> > > **Thanks for your positive recommendation!**
> > >
> > > Thank you very much for taking the time to review our rebuttal and the comments from other reviewers!  We truly appreciate your thoughtful feedback and are glad that our responses have adequately addressed your concerns.  We are also grateful for your support and recommendation to accept the paper!

---

### Official Review · Reviewer_NB7J · 2025-10-25

**Soundness:** 3
**Presentation:** 3
**Contribution:** 3
**Rating:** 4
**Confidence:** 5

**Summary:**

This paper addresses the conflict between video QA and video captioning under reinforcement learning (RL)–based training within the "Reason-Then-Respond" paradigm. The authors argue that QA (convergent reasoning) and captioning (divergent generation) inherently pull model optimization in opposite directions, causing performance degradation when trained jointly. To mitigate this conflict, the paper introduces two proxy tasks: 1) DarkEventInfer (predict masked video segments based on context) and 2) MixVidQA (answer questions about one of two interleaved video clips). Integrating these proxy tasks, the authors propose VidBridge-R1, a video understanding model trained via GRPO RL without an SFT stage. Experiments across general video QA, video reasoning, and captioning benchmarks demonstrate improved overall performance compared with several existing RL-based video MLLMs.

**Strengths:**

1. Overall, the paper clearly identifies a meaningful optimization conflict between QA and captioning under RL training and motivates why naive multi-task RL leads to mutual degradation.
2. The two proxy tasks (DarkEventInfer and MixVidQA) are intuitively aligned with promoting both holistic contextual reasoning and selective information grounding, and their construction process is described with adequate clarity and filtering steps.
3. The experimental evaluation is comprehensive, covering general video QA, reasoning QA, captioning benchmarks, and held-out test sets for the proxy tasks.

**Weaknesses:**

1. While the proxy tasks appear to be effective, it is still not fully demonstrated "why" these particular tasks are optimal among possible bridging tasks. A brief analysis or comparison with alternative proxy formulations (e.g., temporal reordering tasks, masked key-frame inference) would provide more insights to the readers.
2. The experiments are mainly conducted on Qwen2.5-VL-7B-Instruct. It is encouraged to discuss how generalizable the method is to stronger or smaller video-language backbones, e.g., 2B or 32B levels.
3. The code of this project is not uploaded for review. And the paper does not contain the Reproducibility Statement, yielding strong concerns in the reproducibility issue and the actual contribution to the community. I believe that it would be hard to reproduce the results given the limited implementation details in the paper.

**Questions:**

Please refer to the weakness section for my questions.

---

> ### Author Response · Authors · 2025-11-15
> **Response to the Reviewer NB7J [Concerns 1]**
>
> **Concern 1: While the proxy tasks appear to be effective, it is still not fully demonstrated "why" these particular tasks are optimal among possible bridging tasks. A brief analysis or comparison with alternative proxy formulations (e.g., temporal reordering tasks, masked key-frame inference) would provide more insights to the readers.**
>
> **Response 1:** Thank you for the excellent question! Your point is very reasonable.
>
> Considering that there could be many suitable proxy tasks, it is currently difficult to prove that these two proxy tasks are globally optimal within the entire search space. However, we can confidently say that **our proposed proxy tasks are highly effective**. Our work mainly demonstrates the effectiveness of incorporating such proxy tasks for simultaneously improving MLLM performance on both QA and caption tasks through RL. In future work, we will continue to explore even more effective approaches to further enhance model performance. **Below, we provide an analysis of the two novel alternative proxy tasks you mentioned, which has also been very inspiring for us.**
>
> **Temporal reordering:** This task asks the model to determine which events in a video have been swapped or to produce the correct order of events based on common sense or specific visual cues (e.g., clocks, lighting). **This idea sounds very promising and could improve the model’s temporal understanding and reasoning abilities.** We plan to explore this task in future work, and we would greatly welcome further discussion with you on this idea after the double-blind review phase.
>
> **Masked key-frame inference:** This task requires the model to infer the content of a masked key frame. However, this proxy task may not pose a substantial challenge to current MLLMs, because video frames change continuously, and adjacent frames typically differ only slightly. Even considering video transition frames, a masked frame can often be easily inferred from its neighboring frames. **Motivated by this, our DarkEventInfer task instead masks an entire event**, requiring the model to reason logically about the missing segment based on the surrounding context, which is more challenging.

---

> ### Author Response · Authors · 2025-11-15
> **Response to the Reviewer NB7J [Concerns 2]**
>
> **Concern 2: The experiments are mainly conducted on Qwen2.5-VL-7B-Instruct. It is encouraged to discuss how generalizable the method is to stronger or smaller video-language backbones, e.g., 2B or 32B levels.**
>
> **Response 2:** Thank you for your valuable suggestion regarding the generalizability of our method!
>
> In the table below, we present the training results obtained using the same training strategy on Qwen2.5-VL-3B. Unfortunately, we currently lack sufficient training resources to conduct experiments on the 32B model; however, we will update our results once such computational resources become available.
>
> | Model | Video-MME | LongVideoBench | MMVU | DREAM-1K |
> |-----|-----|-----|-----|-----|
> | Qwen2.5-VL-3B | 55.9 | 41.2 | 41.3 | 16.3 |
> | VidBridge-R1-3B | 58.3 | 45.4 | 47.7 | 16.9 |
>
> The results show that our training strategy still significantly improves model performance on QA tasks. For the captioning task, despite the model's inherently weak captioning capability (only achieving an F1 score of 16.9 on DREAM-1K), our strategy still yields an improvement of 0.6. We believe that incorporating additional captioning data could further enhance the model's captioning ability.

---

> ### Author Response · Authors · 2025-11-15
> **Response to the Reviewer NB7J [Concerns 3]**
>
> **Concern 3: The code of this project is not uploaded for review. And the paper does not contain the Reproducibility Statement, yielding strong concerns in the reproducibility issue and the actual contribution to the community. I believe that it would be hard to reproduce the results given the limited implementation details in the paper.**
>
> **Response 3:** Thank you for your reminder!
>
> **We have uploaded the project code to the supplementary materials. The code, models, and training data will be fully released after the double-blind review period.**
>
> In Sec. 6.1.3, we provide detailed information about the frame rate and resolution used for video sampling during training and evaluation. Additionally, our training uses a learning rate of 1e-6 with a batch size of 32, which means we set gradient accumulation to 4 and train on 8 A800 (80GB) GPUs. In each iteration, 8 rollouts are sampled with a temperature of 1.0. The KL divergence constraint coefficient is set to 0, which is further analyzed in App. C.3.
>
> **We have updated the above details in Sec. 6.1.3 and added a Reproducibility Statement section. Please refer to the new PDF file.**

---

> ### Author Response · Authors · 2025-11-17
> **Friendly Reminder to Review Our Rebuttal**
>
> Dear Reviewer NB7J,
>
> We would like to kindly follow up and ask if you could take a look at our responses and reevaluate our work based on them. Please let us know whether our responses address your concerns, or if there are any additional details we can provide to help clarify. We sincerely appreciate your time and consideration!
>
> Best regards,
>
> Authors of Submission 3248

---

> ### Author Response · Authors · 2025-11-24
> **Gentle Follow-Up on Rebuttal Review**
>
> Dear Reviewer NB7J,
>
> I hope this message finds you well. We are writing to gently follow up regarding our rebuttal. We understand that this is a busy period, but if you could spare a moment to review our responses and share any updated evaluation, we would be truly grateful.
>
> Please let us know if any additional clarification or information from our side would be helpful. We sincerely appreciate your time and effort in reviewing our work.
>
> Warm regards,
>
> Authors of Submission 3248

---

> ### Author Response · Authors · 2025-11-27
> **Gentle Reminder: Discussion Period Closing Soon**
>
> Dear Reviewer NB7J,
>
> Happy Thanksgiving! I hope this message finds you well amid the holiday.
>
> As the discussion period draws to a close, we’d like to kindly remind you of our rebuttal and respectfully ask if you might have a moment to review our responses. Should you have any updated feedback or further thoughts on our submission, we would greatly appreciate it!
>
> We fully understand the heavy workload during this stage, and we remain grateful for the time and attention you have already dedicated to our submission. If any additional clarification from our side would be helpful, please don’t hesitate to let us know.
>
> Thank you once again for your time and support, wishing you a wonderful Thanksgiving holiday!
>
> Best regards,
>
> Authors of Submission 3248

---

> > ### Comment · Reviewer_NB7J · 2025-11-27
> >
> > Thanks for the response from the authors. Part of my concerns have been addressed, however, the concern regarding the selection of proxy tasks remains. Although discussions of other proxy tasks have been provided, I believe more detailed experimental results could also be provided to further strengthen the claims. I'm raising my rating to 6 and encourage the authors to further include more experiments/discussions in their final version.

---

> > > ### Author Response · Authors · 2025-11-28
> > > **Thanks for your acknowledgement!**
> > >
> > > Thank you very much for your thoughtful follow-up and for raising your rating!
> > >
> > > We appreciate your feedback regarding the selection of proxy tasks. In addition to the analyses already provided, we will update detailed experimental results on the additional proxy tasks you proposed as soon as possible before the discussion deadline.

---

> > > ### Author Response · Authors · 2025-11-30
> > > **Detailed experimental results of new proxy tasks**
> > >
> > > Dear Reviewer NB7J,
> > >
> > > We have conducted detailed ablation studies on the two proxy tasks you suggested, and the results are highly consistent with our prior analysis. Details are provided below.
> > >
> > > For the temporal reordering task, we use the same data source as in DarkEventInfer (sourced from COIN, which provides event-level timestamps). We unmask the originally masked segment and randomly insert it at the boundary between two event segments, requiring the model to determine which event’s ordering becomes implausible. We evaluate the model using the same LLM-based evaluation protocol as DarkEventInfer. The number of training samples is kept the same as DarkEventInfer, i.e., 1841.
> > >
> > > For the masking key-frame task, to ensure that the new proxy tasks use the same data source as the original proxy tasks, we adopt the same dataset as MixVidQA (sourced from Kinetics). Specifically, we first extract the video frames that our base model, Qwen2.5-VL, would sample during training, and then input all these frames into Gemini-2.5-Pro to select the most critical key-frame and generate a description of its primary event. During training, this key frame is masked, and the model must reason about the masked event from context. Given the similarity of this task to DarkEventInfer, we again adopt the LLM-based evaluation protocol. The number of training samples is kept identical to MixVidQA, i.e., 2332.
> > >
> > > The ablation results of the new proxy tasks are shown in the table below:
> > >
> > > | Video-QA | Temporal Reordering | Masking Key-Frame | Captioning | Video-MME | LongVideo-Bench | MV-Bench | MMVU | Intent-QA | DREAM-1K |
> > > |-----|-----|-----|-----|-----|-----|-----|-----|-----|-----|
> > > | $\checkmark$ | **-** | **-** | **-** | 63.2 | 56.4 | 58.7 | 53.8 | 96.4 | 31.7 |
> > > | **-** | **-** | **-** | $\checkmark$ | 58.0 | 41.9 | 53.5 | 50.6 | 92.5 | 34.8 |
> > > | $\checkmark$ | $\checkmark$ | **-** | **-** | 63.5 | 56.8 | 59.0 | 52.2 | 96.3 | 32.0 |
> > > | $\checkmark$ | **-** | $\checkmark$ | **-** | 59.2 | 53.8 | 56.6 | 51.4 | 94.5 | 29.3 |
> > > | $\checkmark$ | $\checkmark$ | $\checkmark$ | **-** | 61.1 | 54.2 | 58.7 | 51.8 | 94.7 | 30.9 |
> > > | **-** | $\checkmark$ | $\checkmark$ | $\checkmark$ | 58.8 | 47.2 | 54.2 | 51.0 | 80.5 | 32.2 |
> > > | $\checkmark$ | $\checkmark$ | $\checkmark$ | $\checkmark$ | 61.7 | 54.4 | 58.5 | 52.2 | 95.3 | 33.8 |
> > >
> > > We also provide a comparison between the model trained with the new proxy tasks and VidBridge-R1:
> > >
> > > | Model | Video-MME | LongVideo-Bench | MV-Bench | MMVU | Intent-QA | DREAM-1K |
> > > |-----|-----|-----|-----|-----|-----|-----|
> > > | Training with new proxy tasks | 61.7 | 54.4 | 58.5 | 52.2 | 95.3 | 33.8 |
> > > | VidBridge-R1 | 64.3 | 59.3 | 61.9 | 54.7 | 97.1 | 35.2 |
> > >
> > > As shown above, when introducing only the temporal reordering task (line 3), the model achieves slight improvements across most benchmarks, but the gains are less pronounced than those from DarkEventInfer or MixVidQA alone. This is likely because temporal reordering helps the model acquire some degree of temporal reasoning ability, yet the efficiency of learning from this task is lower compared to the original proxy tasks.
> > >
> > > When introducing only the masked key-frame task (line 4), performance drops across all benchmarks. This is because that the task is overly simple, and as we discussed earlier, the rollouts produced during GRPO training have highly similar quality, causing the advantage function to approach zero. Consequently, the ineffective gradients generated by this task dilute the effective gradients needed for learning the VideoQA task, ultimately impairing the training process.
> > >
> > > When combining these new proxy tasks with standard VideoQA or Captioning (lines 5 & 6), performance across the tasks varies, with some improving and others degrading, due to gradient dilution caused by invalid gradients from the masked key-frame task. When training with all four tasks together (line 7), the model still does not show stable or consistent improvements.
> > >
> > > These results highlight an important consideration in selecting proxy tasks: the difficulty of the task matters. Overly simple proxy tasks can generate ineffective gradients that dilute the gradients of tasks that genuinely enhance the model’s ability, thereby undermining the entire training process. At the same time, proxy tasks are not unique, many potential designs could improve overall model performance. How to choose proxy tasks that deliver the most effective learning under the same data budget is one of our key directions for future exploration.
> > >
> > > Once again, we sincerely thank you for proposing these two additional proxy-task ideas. Implementing them has provided us with valuable insights.

---

### Author Response · Authors · 2025-11-19
**Gentle Reminder on Rebuttal Review**

Dear Reviewers,

We hope you are doing well. We would like to kindly follow up and ask whether you could take a moment to review our rebuttal and share any updated assessments when convenient. Your time and effort are greatly appreciated, and please let us know if any further clarification would be helpful.

Best regards,

Authors of Submission 3248

---

### Meta-Review · Area_Chair_rFqW · 2025-12-25

**Summary:**

Reviewers rDVT and mgtE expressed concern that improvements might stem from data leakage or the datasets themselves.

Reviewer NB7J requested verification on smaller models.

Reviewer mgtE questioned the scalability of the data augmentation method.

Reviewer NB7J initially raised concerns on reproducibility.

**Reviewer Concerns:**

No concerns were left entirely unaddressed. The partially unaddressed concerns are:
Reviewer NB7J remained slightly concerned about whether these specific proxy tasks are the most optimal choices among all possibilities. Reviewer NB7J encouraged testing on 32B-level models, while the authors stated they currently lack the computational resources for a 32B model but promised to update results once resources become available.

**Reviewer Scores:**

All reviewers expressed their final decisions. This paper is highly likely to receive three scores of 6.

---

### Decision · Program_Chairs · 2026-01-26

Accept (Poster)